# Stratospheric ozone and QBO interaction with the tropical troposphere on intraseasonal and interannual time-scales: a normal mode perspective

Breno Raphaldini[1], Andre Seiji Watake Teruya[1], Pedro Leite da Silva Dias[1], Lucas Massaroppe[1], and Daniel Yasumasa Takahashi[2]

[1]Department of Atmospheric Sciences, University of São Paulo
[2]Instituto do Cérebro, Federal University of Rio Grande do Norte

**Correspondence:** Breno Raphaldini (brenorfs@gmail.com)

**Abstract.** The Madden Julian Oscillation (MJO) is the main controller of the weather in the tropics on intraseasonal time-scales and recent research provides evidences that the Quasi-Biennial Oscillation (QBO) influences the MJO interannual variability. However the physical mechanisms behind this interaction are not completely understood. Recent studies on the normal mode structure of the MJO indicates the contribution of global-scale Kelvin and Rossby waves. In this study we test whether these MJO-related normal modes are affected by the QBO and stratospheric ozone. The Partial Directed Coherence method was used and enabled us to probe the direction and frequency of the interactions. It was found that equatorial stratospheric ozone and stratospheric zonal winds are connected with the MJO at periods of 1-2 months and 1.5-2.5 years. We explore the role of normal mode interactions behind the stratosphere-troposphere coupling by performing a linear regression between the MJO/QBO indices and the amplitudes of the normal modes of the atmosphere obtained by projections on a normal mode basis using ERA-Interim reanalysis data. The MJO is dominated by symmetric Rossby modes but is also influenced by Kelvin and asymmetric Rossby modes. The QBO is mostly explained by westward propagating inertio gravity waves and asymmetric Rossby waves. We explore the previous results by identifying interactions between those modes and between the modes and the ozone concentration. In particular, westward inertio-gravity waves, associated with the QBO, influence the MJO on interannual time-scales. MJO related modes such as the Kelvin wave and the Rossby wave with symmetric wind structure with respect with the equator are shown to have significantly different dynamics during MJO events depending on the phase of the QBO.

## 1 Introduction

The Madden Julian Oscillation (MJO) and the Quasi-Biennial Oscillation (QBO) are two of the main elements of the atmospheric low frequency variability in the tropics. The MJO acts on intraseasonal time-scales on the troposphere and impacts the tropical monsoons and with global impacts (*Zhang*, 2005). The QBO manifests in the tropical stratosphere as a reversal of the

zonal winds with descending cycles with mean period of 28 months also with important impacts on the Global circulation of the atmosphere (*Holton & Tan*, 1980). Both are important players for the Earth system's weather and climate. Careful examination of causal relationships between such processes and the physical mechanisms behind their interaction are active topics of research in recent years (*Zhang et al.*, 2018).

5      The stratosphere can act as a mediator between solar forcing and the climate variability of the troposphere. It is conjectured that stratospheric influence on the troposphere exists via the so-called top-down mechanism (*Gray et al.*, 2010). According to this hypothesis, stratospheric ozone absorbs ultraviolet solar (UV) radiation releasing heat. This heat then generates temperature and wind perturbations in the stratosphere that might then induce a tropospheric response through downward energy transport. However, details of the physical mechanisms through which stratospheric signals could propagate down to the troposphere are 10 not completely understood.

     Stratospheric control of tropospheric phenomena in mid to high latitudes was addressed in several papers. For instance, *Baldwin et al.* (2010) highlights the polar vortex as an important example of such control. Another example is that of stratospheric impacts on tropospheric upper level jets and storm tracks as seen in *Kidston et al.* (2015). *Yoo & Son* (2016) showed that the MJO is sensitive to the QBO phase in the annual timescale, concluding that including QBO information improves the MJO 15 predictability (*Marshall et al.*, 2017; *Son et al.*, 2017). *Densmore et al.* (2019) attributes differences on the QBO-MJO interaction to the QBO phase to differences in the static stability of the upper troposphere/lower stratosphere, leading to changes in the excitation of MJO-related disturbances. *Hendon & Abhik* (2018) associated the increased predictability and intensity of the MJO during the boreal winter and QBO easterly phase with differences in the vertical structure of the MJO, depending on the QBO phase. The problem of MJO-QBO connection is however still not well understood from the perspective of the underlying 20 physical mechanism nor well represented in numerical models as pointed out recently in *Kim et al.* (2020).

     The study of QBO effects on the MJO gained a lot of interest in the last few years, since new evidence pointed out this connection (*Yoo & Son*, 2016). Since then several articles explored both the physical mechanisms behind this interaction as well as consequences for weather and climate. One of the main factors that plays a role in the QBO-MJO connection is the difference in the static stability at the Tropopause region depending on the phase of the QBO (*Nishimoto & Yoden*, 2017). 25 *Hendon & Abhik* (2018) suggests that negative temperature anomalies at the tropopause region during the easterly QBO phase act to destabilize the upper troposphere in phase with MJO associated convection, thus reinforcing the MJO event . Alternative mechanisms that could contribute to this stratosphere-troposphere connection include the downward reflection of planetary waves (*Lu et. al*, 2017) and effects on tropospheric Rossby wave-guides and teleconnection patterns (). Here we investigate a different class of mechanism, namely the role of wave interaction. Nonlinear wave interaction is believed to have a role in 30 the initiation of an MJO event though the interaction between the tropics and extra-tropics (see section 6.4 ()). This interaction takes place by by the coupling between equatorially confined modes, the baroclinic Rossby waves, and non-confined modes, the barotropic Rossby waves. Inspired by this type of mechanism we investigate whether the interaction between QBO-related modes with MJO-related modes could have a role in the MJO-QBO connection.

     Recent studies have given a normal mode description of the MJO (*Zagar & Franzke*, 2015; *Kitsios et al.*, 2019). These 35 studies concluded that the MJO can be described as global scale baroclinic Rossby and Kelvin waves. The same approach was

used to study the conditions that lead to the 2016 QBO disruption (*Raphaldini et al.*, 2020). In this context a natural question arises: what is the role of these normal modes in the MJO interaction with the stratosphere? In particular, how do these modes interact with QBO-related modes?

In this article, we study the interactions between the stratosphere and the tropical troposphere, with particular emphasis on the MJO. A time series analysis causality method, Partial Directed Coherence (PDC), (*Baccala & Sameshima*, 2001) was used. We determine whether equatorial ozone, equatorial stratospheric zonal winds and tropospheric fields interact and how this interaction occurs, including information on directional interaction. Our analysis is based on daily data of stratospheric zonal wind, ozone concentration, and the unfiltered (on the intreseasonal timescale) MJO index from 1979 to 2015. We obtained stratospheric zonal wind and ozone concentration from ERA-Interim reanalysis data (*Dee et al.*, 2011) from the European Centre for Medium-Range Weather Forecasts. Zonal wind at 30 hPa level was averaged in an equatorial belt from $-15^o$ to $15^o$ latitude for all longitudes, which is a reasonable choice to represent the QBO (*Nappo*, 2013). Ozone data were averaged from $-20$ to $20$ in latitude and integrated over all levels from $100$ to $0.1 hPa$. MJO data was obtained from the daily MJO index RMM (*Wheeler & Hendon*, 2004). The MJO index is presented in a polar coordinate diagram with two time-series, amplitude and phase. The amplitude of the MJO index is defined as the sum of the squares of the first two empirical orthogonal functions (EOFs) of combined pressure fields at $200$ and $850 hPa$ and outgoing long wave radiation data in the tropics (RMM1 and RMM2). An equivalent way to represent the MJO index (a complex number) is to use two real variables that correspond to the two first components. In order to use minimal mathematical operations with the original EOF time-series we choose the last representation.

To resolve the spectrum of the different time-scales, time-scale separation was applied to the data. We split the data into a fast time-scale (periods shorter than one year), and a slow time-scale (periods greater than one year). This was done by performing a resampling procedure on the data with a ten-day rate for the "fast" time-scale. A six-month window was applied for the "slow" time-scale.

The causality between the QBO, tropical stratospheric ozone and the MJO, was studied using the PDC method. PDC corresponds, roughly, to a frequency domain counterpart of the Granger Causality test (*Baccala & Sameshima*, 2001), with the additional advantage of providing information on the specific frequencies at which the causality occurs.

We seek for normal modes that might contribute to the interactions between stratospheric and tropospheric phenomena by performing a linear regression with the MJO indices and stratospheric zonal winds. We then perform the PDC analysis with the time series for the energies associated with each of the Hough modes responsible for the MJO dynamics (as in *Zagar et al.* (2015)) and of the stratospheric zonal wind. The results indicate that the interaction of internal westward gravity waves, responsible for the QBO and Kelvin, and Rossby waves associated with the MJO, partially explain the stratospheric influences on the MJO.

## 2 Methods

### 2.1 Granger Causality

The concept of causality is a central question in science. One possible definition of causality related to the predictability of two or more distinct processes was introduced in *Granger* (1969) and is currently known as Granger causality in the literature. The main advantage is the ability to pinpoint the direction of interaction, unlike other measures such as coherence, correlation, partial coherence and partial correlation. The following definition is specific to trivariate time series but is readily generalizable to an arbitrary number of time series.

Consider a vector-valued signal $\mathbf{X}(t) = [X_1(t), X_2(t), X_3(t)]^\top$ where the supersprict $^\top$ indicates the transpose of a vector and $\mathbf{X}(t)$ is assumed to have a vector autoregressive representation of order $p$ (hereafter referred as VAR($p$))

$$
\begin{bmatrix} X_1(t) \\ X_2(t) \\ X_3(t) \end{bmatrix} = \sum_{k=1}^{p} \begin{bmatrix} a_{11}(k) & a_{12}(k) & a_{13}(k) \\ a_{21}(k) & a_{22}(k) & a_{23}(k) \\ a_{31}(k) & a_{32}(k) & a_{33}(k) \end{bmatrix} \begin{bmatrix} X_1(t-k) \\ X_2(t-k) \\ X_3(t-k) \end{bmatrix} + \begin{bmatrix} \epsilon_1(t) \\ \epsilon_2(t) \\ \epsilon_3(t) \end{bmatrix}, \tag{1}
$$

where $a_{ij}(k)$ are the VAR($p$) coefficients representing the $k-$th lagged influence of the $j-$th component of the signal on the $i-$th component and $t$ denotes the time variable. The innovations processes (the random component) $\epsilon_i(t)$ have zero mean and covariance matrix $\mathbf{C} = [\sigma_{ij}]$, such that $Cov(\epsilon_i(t), \epsilon_j(s)) = 0$ for $t \neq s$ and for all $i, j \in \{1, 2, 3\}$.

It is enough to say that $X_j(t)$ Granger causes $X_i(t)$ for $i \neq j$ if $a_{ij}(k) \neq 0$, with statistical significance, for some lag $k = 1, \ldots, p$. Thus, the absence of Granger causality from $X_1(t)$ to $X_2(t)$ implies that $X_1(t)$ does not help to predict $X_2(t)$, once the past of $X_2(t)$ and $X_3(t)$ are considered.

In practice, given a trivariate time series $\mathbf{X}(t)$ of length $n$, we estimate the VAR($p$) model from the data and test for $a_{ij}(k)$ nullity. More precisely, the idea is verify the null hypothesis

$$
\mathcal{H}_0 : \ a_{ij}(k) = 0, \ k = 1, \ldots, p, \tag{2}
$$

against

$$
\mathcal{H}_1 : \ \text{there exist } k \in \{1, \cdots, p\}, \ \text{such that } a_{ij}(k) \neq 0. \tag{3}
$$

Therefore, we can say that the $j-$th component of the time series causes the $i-$th component in the sense of Granger if the past of the $j-$th component helps to predict the future of the $i-$th component. We have used the MATLAB Toolbox (free) implementation of the VAR($p$) and Granger causality estimators implementations from *Sameshima et al.* (2015), available at http://www.lcs.poli.usp.br/~baccala/pdc.

### 2.2 Partial directed coherence

Partial Directed Coherence (PDC) is an extension of the concept of Granger causality to the frequency domain, as a measure of information flow. Thus, PDC incorporates advantages of the Granger causality and of the classical coherence methods with

the additional advantage that it can be generalized to more than two time series enabling to explicitly pinpoint the directed information flow from mere indirect interactions, (*Baccala & Sameshima*, 2001; *Takahashi et al.*, 2007, 2010). PDC has been successfully applied in complex systems as neurocience (*Baccala & Sameshima*, 2001; *Schelter et al.*, 2006) and economics (*Hui & Chen*, 2012). PDC was also used to detect the causality between the El Niño Southern Oscillation and the monsoons and also in the sea-air interaction in the South Atlantic Convergence Zone (*Tribassi et. al*, 2017).

Again, consider a trivariate time series $\mathbf{X}(t) = [X_1(t), X_2(t), X_3(t)]^\top$ with a VAR($p$) representation defined in (1), let

$$\bar{A}_{kl}(\nu) = \delta_{kl} - \sum_{s=1}^{p} a_{kl}(s)\mathrm{e}^{-\mathrm{i}2\pi\nu\mathrm{s}}, \tag{4}$$

where $\delta_{kl}$ is the Kronecker delta symbol, $\mathrm{i}^2 = -1$, $\nu$ the Fourier frequency (in Hertz), $s$ the time (in seconds). Here we use the more general PDC definition, the information-Partial Directed Coherence ($_i$PDC), which is closely related to information theory. It has been shown that $_i$PDC corresponds to the information flow (in Shannon's sense) between different signals (*Baccala et al.*, 2013). Therefore the information flow, $_i$PDC, from $X_j(t)$ to $X_i(t)$ in a specific frequency $\nu$, is given by

$$_i\mathrm{PDC}_{i\leftarrow j}(\nu) := {}_\iota\pi_{ij}(\nu) = \frac{\bar{A}_{ij}(\nu)/\sqrt{\sigma_{ij}}}{\sqrt{\bar{\mathbf{a}}_j^{\mathsf{H}}(\nu)\mathbf{C}^{-1}\bar{\mathbf{a}}_j(\nu)}}, \tag{5}$$

where $\bar{\mathbf{a}}_j(\nu)$ is the $j$−th column of the matrix with coefficients $\bar{A}_{kl}(\nu)$, and $\bar{\mathbf{a}}_j^{\mathsf{H}}(\nu)$ denotes its Hermitian transpose.

Note that there is a duality between the Granger causality and PDC, as demonstrated in *Sameshima et al.* (2015). Therefore the nullity of $_\iota\pi_{ij}(\nu)$ corresponds to the absence of connection (similarly to the aforementioned Granger causality condition), which, in the PDC case, also has a rigorous and well-defined statistical criterion for the null hypothesis test (*Baccala et al.*, 2013). Confidence intervals for the PDC analysis are explicitly calculated as the statistics of the PDC coefficients, $_\iota\pi_{ij}(\nu)$, is asymptotically Gaussian (at the limit of a large number of data points). For a proof of this theorem and more information on confidence intervals for PDC see *Baccala et al.* (2013) and *Takahashi et al.* (2007). To estimate the $_i$PDC from the data, the first step is to obtain the vector autoregressive model, which is estimated through the Hannan-Quinn criterion in this paper and substitute the estimated coefficients in Eq.(3). The implemented test statistics are described in *Baccala et al.* (2013), and we used the computations of $_i$PDC generated from AsympPDC Package version 3.0 MATLAB Toolbox freely available as mentioned before. A detailed example showing how to interpret the PDC plots is given in the supplementary material (see figure S1).

The partial directed coherence as well as Granger causality related quantities are linear measures and a natural question is whether these methods are able to capture the interaction between signals that arise from nonlinear problems. There are several publications addressing this question such as possible nonlinear extension of this technique (*Massaroppe & Baccala*, 2015; *Wahl et al.*, 2016) and the introduction of other techniques that are intrinsically nonlinear in nature, based on time lagged embedding, such as *Sugihara et al.* (2012), or based on the concept of Markov partitions, such as *Bianco-Martinez et al.* (2018). *Sugihara et al.* (2012) gives an example in which Granger based techniques perform poorly. Here we argue that although PDC does not capture all kinds of nonlinear coupling between time scales especially with more intermittent/non-Gaussian behavior, it certainly captures certain kinds of nonlinear interactions. As proved in *Takahashi et al.* (2010) there is an equivalence between

the concepts of mutual information rate that would account for all information flow between two or more signals and PDC, in the case of Gaussian processes. In the general non-Gaussian case bounds are given for the difference of the mutual information rate estimated by PDC and the actual mutual information rate, meaning that even if the signals are nonlinear and non-Gaussian PDC is still able to capture part of the information flow between the signals.

The main advantage of PDC and Granger causality is that it is theoretically related to the mutual information rate (MIR) between signals (*Takahashi et al.*, 2010). Information-theoretic quantities are usually costly to estimate directly from time-series since it relies on the estimation of multi-dimensional probability distributions. As proved in Takahashi et. al 2010, PDC is a Gaussian approximation to the MIR. This means that if the time-series are stationary and Gaussian, PDC provides an exact estimate for the MIR, when the time-series are not Gaussian (possibly due to underlying nonlinearities) the PDC will capture part but not all of the information flow between the time-series. There are many "causality" estimation methods in the literature, all of them with some advantages and drawbacks. Among the several causality detection methods the Convergent-Cross Mapping (CCM) method is proposed as a method that is capable of capturing couplings in highly-nonlinear settings since it relies on phase-space embedding procedures. CCM. However, it comes with a few drawbacks that would require more in-depth investigation before we could apply it in the present setting, namely: (1) CCM is a bi-variate measure. Granger causality and PDC are genuinely multivariate measures. (2) CCM may lead to wrong or misleading results when moderate to high levels of noise are present (see (*Monster*, 2017)). Granger causality and PDC are designed to work for signals with stochasticity. (3) CCM does not have an automated way to decide the optimal lag between time series. Granger causality and PDC are based on autoregressive process in which order estimation is well studied. (4) There are no theoretical guarantees for the statistical properties of CCM. Both PDC and Granger causality are at very well studied measures in which there are thousands of articles applying it and we understand well their statistical properties (*Lutkepohl*, 2005; *Takahashi et al.*, 2007).

Finally, although PDC is a stochastic linear method, it correctly reconstruct the topology of networks of nonlinear oscillators, see *Winterhalder et al.* (2007), Moreover, it has been successfully and extensively used to infer information flow in highly nonlinear time-series data in neuroscience (*Sato et al.*, 2009). The fact that PDC can detect nonlinear interactions is not difficult to understand, given that linear regression also can see nonlinear interaction unless the nonlinearity is highly non-monotonic.

## 2.3 PDC statistics

The PDC is a function of the coefficients of vector autoregressive model. Given that the coefficients are asymptotically jointly normally distributed, we can use the delta method (*Serfling*, 1980) to obtain analytically the asymptotic statistics for PDC. After an algebraic computation we can show that PDC at frequency lambda is distributed asymptotically (under the null hypothesis of zero PDC) as the weighted sum of two chi-square with one degree of freedom ((*Takahashi et al.*, 2007; *Baccala et al.*, 2013)). Therefore, we can use the asymptotic distribution to calculate the p-value. More specifically, let $_\iota\hat{\pi}_{ij}(\nu)$ be the estimator of $_\iota\pi_{ij}(\nu)$ for a time series of length $n$. We have the following convergence in distribution.

$$n\,\bar{\mathbf{a}}_j^{\mathsf{H}}(\nu)\mathbf{C}^{-1}\bar{\mathbf{a}}_j(\nu)\left(|_\iota\hat{\pi}_{ij}(\nu)|^2 - |_\iota\pi_{ij}(\nu)|^2\right) \xrightarrow{d} l_1 Y_1 + l_2 Y_2, \tag{6}$$

where $Y_1$ and $Y_2$ are independent $\chi^2_1$ distributed random variables and $l_1$ and $l_2$ are weights that can be estimated from the data. For details of the derivation, we refer to *Takahashi et al.* (2010). The significance level used in the article for PDC is the frequency-wise value as it is the standard for frequency domain analysis given the high correlation between the point estimates for neighboring frequencies (*Huybers & Curry*, 2006; *Came*, 2007)).

## 2.4 Normal mode decomposition

Based on the methodology of *Kasahara & Puri* (1981), *Zagar et al.* (2015) introduced a software for the projecting atmospheric fields from reanalysis onto the normal modes of the hydrostatic primitive equations on the sphere. For a vector valued function $\mathbf{X} = [u, v, h]^\top$, where $u(\lambda, \phi, z)$ is the zonal velocity field, $v(\lambda, \phi, z)$ is the meridional velocity field, $h(\lambda, \phi, z)$ is the modified geopotential height. A separation of variables is then performed and the state vector $\mathbf{X}$ is represented as a series of horizontal and vertical structure functions, which in discrete form is

$$\mathbf{X}(\lambda, \phi, z) = \sum_{m=1}^{M} \mathbf{S}_m \mathbf{X}_m(\lambda, \phi) G_m(z), \tag{7}$$

where $\mathbf{X}_m$ is the horizontal structure vector function, $G_m$ is the vertical structure function and $\mathbf{S}_m$ is a square matrix defined as

$$\mathbf{S}_m = \begin{bmatrix} \sqrt{gD_m} & 0 & 0 \\ 0 & \sqrt{gD_m} & 0 \\ 0 & 0 & D_m \end{bmatrix},$$

where $g$ is Earth's gravity and $D_m$ equivalent depth of the $m-$th vertical mode. The horizontal fields $\mathbf{X}_m$, on the other hand, are expanded in Hough harmonics as

$$\mathbf{X}_m(\lambda, \phi) = \sum_{n=1}^{N} \sum_{k=-K}^{K} \chi_{m,n,k} \mathbf{H}_{m,n,k}(\lambda, \phi), \tag{8}$$

where $\mathbf{H}_{m,n,k}$ are the eigenfunctions of the Laplace's tidal equation considering zonal periodicity and regularity at the poles as boundary conditions (*Longuet-Higgins & Selwyn*, 1968). The expansion coefficients $\chi_{m,n,k}$ are obtained as

$$\chi_{m,n,k} = \frac{1}{2\pi} \int_0^{2\pi} \int_{-1}^{1} \mathbf{X}_m(\lambda, \phi) \cdot [\mathbf{H}_{m,n,k}(\lambda, \phi)]^* \, d\mu d\lambda, \tag{9}$$

with $\mu = \sin(\phi)$ and the superscript $^*$ indicates the complex conjugate. Details of the procedures for obtaining the amplitudes $\chi_{m,n,k}$ from the data is described in *Zagar et al.* (2015). The MODES software then provides the amplitudes $\chi_{m,n,k}$ given input time scales of reanalysis data. *Zagar & Franzke* (2015) proposed a procedure to decompose the MJO into the contributions of each normal mode by performing a linear regression between the MJO time series and the mode-amplitude time series

$$\mathcal{R}_{m,n,k} = \frac{1}{N-1} \sum_{t=1}^{N} \frac{(\chi_{m,n,k}(t) - \mathbf{E}[\chi_{m,n,k}(t)])(Y(t) - \mathbf{E}[Y(t)])}{Var[Y(t)]} \tag{10}$$

where $\chi_{m,n,k}(t)$ is the Hough expansion coefficient (9) for a time instant $t$, $Y(t)$ is the MJO index time series and $\mathbf{E}[Y(t)]$ and $Var[Y(t)]$ are the respective expectation and variance, respectively.

From the time series of the amplitudes of the normal mode functions we compute the energy within a group of modes, consisting of the sum of the squares of their amplitudes weighted by their equivalent depths $D_m$:

$$E(t) = \frac{1}{2} \sum_{m=M_0}^{\overline{M}} g D_m \sum_{k=0}^{\overline{K}} \sum_{n=N_0}^{\overline{N}} \left( [\mathcal{X}_{kmn}](t)[\mathcal{X}_{kmn}]^*(t) \right) \tag{11}$$

where $\overline{M} = 43, \overline{K} = 32$ and $\overline{N}$ are wavenumber truncations, throughout the text we select different $N$ to represent different modes (Kelvin, Rossby, westward inertio-gravity...).

## 3  Statistical analysis: QBO-MJO-Ozone interaction

Time-series of the stratospheric zonal wind at 30 Mb, equatorial ozone concentration in the stratosphere and the RMM index are presented in Figure 1. The autoregressive fitting of the time series were found to be well-represented by the, passing the Portmanteau test (*Lutkepohl*, 2005). The PDC analysis for the fast (interannual) timescale, Figure 2, indicates that there is a statistically significant interaction between the stratospheric mean zonal wind and the MJO and between tropical stratospheric ozone and the MJO, results here are presented only for RMM1 (RMM2 yield similar results). Concerning the influence of the stratospheric variables on the MJO, tropical stratospheric ozone is shown to have a significant causality (in the Granger sense) on the MJO indices, influencing RMM1 during periods of around one month, corresponding to the higher frequency range of a MJO cycle,. The periods when ozone influences RMM1 and RMM2 show, by the definition of Granger Causality, that information on ozone should improve the MJO predictability.

In order to investigate the interaction between the stratospheric variables and the MJO index we performed a $6-$ month re sampling procedure. Results are presented in Fig. 3. Ozone is found to significantly influence the MJO, as can be seen in Figure 2, on the annual time-scale for RMM2, possibly due to the annual cycle, and on the time-scale of $1.6 - 2.1$ years, possibly associated with the QBO. Both RMM indices are found to be significantly affected at frequencies with a peak at 11 years, which is a strong indication of the effect of the solar cycle on the MJO, through ozone, which could explain the solar cycle related monsoon variability (*VanLoon & Meehl*, 2012), see also *Hood* (2018) for evidence of the impact of the solar variability on the MJO. Interactions that are significant are found from ozone to the MJO in a period ranging from one to two years, possibly as a combination of effects of the annual cycle and the QBO, corroborating the recent results in the literature (*Marshall et al.*, 2017; *Son et al.*, 2017; *Yoo & Son*, 2016).

## 4  Modal decomposition and wave interactions

Several studies point out to the role of the interaction of waves with different vertical structure in the dynamics of the MJO. For instance, *Majda & Biello* (2003) study the interaction of barotropic and baroclinic Rossby waves in the interaction of the

tropics and extra-tropics since barotropic waves are not equatorially confined as the baroclinic ones. *Raupp et al.* (2008) further explores this mechanism in the initiation of the MJO.A Similar mechanism could in principle play a role in stratospheric-tropospheric interactions, with modes with dominant energy in the stratosphere interacting with modes that have more energy in the troposphere. We, therefore, aim to test such a hypothesis.

We initially perform a linear regression analysis between the time series associated with the MJO indices and to the stratospheric zonal wind representative of the QBO, aiming to find which normal modes best represent such oscillations. This analysis was introduced by *Zagar et al.* (2015) in a normal mode decomposition of the MJO. *Zagar & Franzke* (2015) showed that the dominant modes in the decomposition are the symmetric Rossby mode (with the largest contribution coming from the Rossby mode with meridional index 1, denoted by RSSY1 ), as well as Kelvin waves (KW). Both Kelvin and Rossby modes

have larger regression coefficient for the vertical mode indices 5-9, which have a first baroclinic structure in the troposphere. We performed a similar analysis with the daily time-series of equatorial zonal wind at 30 hPa which is dominated by the QBO. We find that the dominant modes in our regression analysis are westward propagating gravity waves (WIG) and the first asymmetric Rossby modes (meridional index 2, denoted by RWASY1), we refer to (*Raphaldini et al.*, 2020) for details on the normal mode decomposition of the QBO.

We seek for interactions between the MJO and QBO normal modes. In order to do so, we calculate the time-series of the energy associated with each of the modes (i.e. a weighted sum of the square of absolute value of each of the modes). We begin by describing the interaction between modes associated with the MJO and to the QBO and tropical stratospheric ozone forcing on sub-annual time scales. Due to the large number of variables we split the analysis into three sets, each containing all the "stratospheric variables" against one of the variables associated with the MJO. Since the most important interactions between

QBO modes and MJO modes are through the QBO-related WIG waves, we restrict the analysis to these modes.

In 5 we present the PDC analysis of the interaction of Kelvin wave vs. westward inertio-gravity wave vs. stratospheric ozone vs. asymmetric Rossby wave, the first three variables associated with stratospheric phenomena and the last one associated with the MJO. We observe that the ozone forcing acts directly on the MJO related Kelvin waves, most notably on intraseasonal time-scales, with a peak around 50 days. The influence of ozone on this mode is also relevant on a semi-annual and annual

time-scale both associated to the annual cycle. WIG waves are found to influence the Kelvin waves on the time-scale of 30 days, while asymmetric Rossby waves are found to influence the Kelvin waves on time-scales from around 50 days to the semi-annual and annual time-scales. We find a feedback from the Kelvin wave to the stratospheric-related variables on intraseasonal, semi-annual and annual time-scales.

Finally, we perform the PDC analysis of the interaction between symmetric Rossby wave (the dominant mode on the MJO

decomposition), asymmetric Rossby wave,WIG wave and stratospheric ozone on the fast time-scale. The corresponding PDC plot is presented in 4. The influence of stratospheric ozone on symmetric Rossby waves has peaks at 40 days, 60 days and on a semi-annual time-scale. The influence of the modes associated to the stratospheric zonal wind on the MJO-related Rossby mode seems to be significant throughout the entire intraseasonal time-scale range, most notably around $30 - 40$ days, as well as on semi-annual and annual time-scales. Similarly to the previous cases, the feedback of the MJO-related mode to the

stratospheric-related variables takes place on intraseasonal, semi-annual and annual time-scales.

We proceed by analyzing the PDC between the modes associated with stratospheric zonal wind and stratospheric ozone vs MJO-related modes on slow time-scales (annual-decadal time-scales). Most importantly, we search for stratospheric influences on MJO on decadal and biennial time-scales. The analysis of the interaction between Kelvin waves, associated with the MJO and tropical stratospheric ozone is presented in Fig.6. It shows that there is a significant causality from ozone to Kelvin waves on a decadal time-scale. Given that both spectra have a peak on the decadal time-scale we can say that the ozone, which is directly influenced by the solar variability, has a peak directly associated with the solar-cycle and the peak on the Kelvin wave spectrum is at least partially explained by the influence of the ozone on it. Kelvin waves on the other hand influence the ozone on annual time-scales, probably due to the annual cycle. The analysis of the interaction between gravity waves associated with the stratospheric zonal wind and the MJO-related Kelvin waves is presented in Fig. 7. We found an important influence of the westward inertio-gravity waves on the Kelvin waves on biennial time-scales and on decadal time-scales. The first one is clearly associated with the biennial peak on the inertio-gravity wave spectrum which is a product of the quasi-biennial oscillation and might be associated to the results of *Yoo & Son* (2016) and subsequent articles on the relationship between the QBO and the MJO. The PDC peak on the decadal time-scale is possibly associated with the solar cycle and the gravity modes are forced by the ozone 9. Since we do not find spectral peaks on this range, we suspect that this is related to the nearest peak, which is annual. A strong causality is also found on a decadal time-scale, again probably due to the solar cycle. The influence of WIG modes on the MJO related Rossby modes is presented in 8, showing a influence of WIG modes on Rossby modes on annual and biennial timescales.

## 4.1 Evolution of MJO normal modes

Previous studies point out to different MJO behaviour depending on the phase of the QBO (east or west) (*Yoo & Son*, 2016), it is therefore important to examine how and if these differences manifest on the MJO-related normal modes. In order to do so we follow the methodology used in *Franzke et al.* (2019) to study Northern hemisphere extra-tropical response of the MJO using normal mode decomposition. We construct composites presenting velocity and pressure fields associated to MJO-normal modes for each phase of the MJO. In order to exclude periods without MJO events we include in our analysis only days in which $(RMM_1^2 + RMM_2^2 > 1)$. We then divide the MJO events in 8 phases depending on the phase of the MJO $\phi = arctg(RMM_2/RMM1)$. For which QBO (positive or negative) state and for which MJO phase (i=1,2,...,8) we calculated the mean velocity and pressure fields associated with ROT and Kelvin modes at 200 Mb.

Figures 10 and 11 display, respectively, the composites associated with the reconstructions of velocity and geopotential height fields associated with ROT modes for each of the 8 MJO phases with positive stratospheric zonal wind at 30 mb (SZW30+) and negative (SZW30+). In order to compare both composites we compute the difference between SZW30+ and SZW30- of each field for each MJO phase. This is displayed in figure 12. We notice that for phases 1-3 the difference (of the geopotential height fields represented by the hatched region) is statistically significant for almost the entire domain. For phase 4 the fields are more similar with small regions with significant difference, associated with Rossby double vortices. Between phases 5-8 the areas with significant difference become larger again.

Figures 13 and 14 display respectively the composites associated with the reconstructions of velocity and geopotential height fields associated with the Kelvin mode for each of the 8 MJO phases with positive stratospheric zonal wind at 30 mb (SZW30+) and negative (SZW30+). In order to compare both composites we compute the difference between SZW30+ and SZW30- of each field for each MJO phase. This is displayed in figure 15. We notice that for phases 1-3 the difference (of the geopotential height fields represented by the hatched region) is statistically significant for almost the entire domain. Unlike in the case of ROT modes, for the Kelvin modes the distribution of statistically significant difference is more even throughout a MJO cycle with a larger area on phase 2 and more similar fields on phase 4. It is possible to notice a propagation pattern with negative geopotential height anomaly beginning at phase 4 and ending at phase 7.

## 5 Final remarks

The PDC results show strong coupling between tropical ozone, stratospheric zonal wind and the MJO. Most notable are the effects of tropical stratospheric winds and ozone influencing the MJO on both intra- and interannual time-scales. The PDC analysis shows that the tropical stratospheric ozone influences the MJO in periods of $30 - 60$ days and $1.5 - 2.5$ years. The first period agrees with the MJO period range, suggesting that stratospheric ozone may play a role in the MJO dynamics. The second roughly agrees with the QBO period and the third suggests a solar cycle influence on the MJO. Stratospheric zonal winds also influence the MJO during periods that fall into the QBO period range, in agreement with the recent results of *Yoo & Son* (2016), who showed that there is an interannual variability in the MJO amplitude that depends on the QBO phase. Marshall (2016) also shows that the QBO explains up to $40\%$ of the MJO interannual variability in the boreal winter (also see *Son et al.* (2017)).

By the definition of Granger causality, one signal causes a second signal if the information of the first helps to predict the future of the other, after taking into account the past of the second signal. In this sense, we confirm the results of the recent studies cited above. We also show that tropical stratospheric ozone also improves the MJO predictability on interannual and decadal time-scales. The periods of interaction suggest that the QBO might be an important process in troposphere/stratosphere coupling through MJO. This conclusion agrees with numerical studies such as that of *Meehl et al.* (2009), stressing the importance of a realistic QBO in coupled troposphere-stratosphere models. We note that ozone influences the MJO on the intraseasonal time-scales, raising the possibility of tropical stratospheric ozone fluctuations contributing to the initiation of the MJO cycle. On the decadal time-scale, ozone and QBO are modulated by solar activity and ozone was shown to have important impacts on the MJO in this time-scale. There is strong evidence in the literature for the solar cycle impact on the Asian monsoons from both instrumental observations and palaeoclimatic reconstruction, with the rainfall rate in the Indian subcontinent increasing by up to $20\%$ during the solar maximum (*VanLoon & Meehl*, 2012). Since monsoons are linked to the MJO, especially in the Indian region where the MJO signal is strongest, it would be natural to hypothesize that the MJO is a mediator between solar variability and monsoons.

It was also found that the MJO can affect stratospheric ozone, a possible mechanism for this being the impact of deep convection on the tropopause height (*Tian et al.*, 2007). Another interesting question is whether the relationship between the MJO and the QBO is affected by the recent anomalous behavior of the QBO (*Osprey et al.*, 2016; *Raphaldini et al.*, 2020).

As for physical mechanisms that could link stratospheric heating, driven by solar UV forcing, and tropical convection, investigation on tropopause changes caused by ozone absorption is a possible candidate. *Kang et al.* (2011) suggested a polar latitudes mechanism associated with changes of wave momentum flux due to ozone depletion associated with the ozone hole. Although this mechanism was proposed for high latitudes, it would be interesting to investigate whether it can be extended to the tropics and to ozone changes due to the annual and solar cycles. Recently, *Lu et. al* (2017) suggested that changes in the wave-guides of planetary waves in the stratosphere, caused by solar forcing changes in the mean flow of the stratosphere, might cause downward planetary wave reflection in high solar activity conditions.

We performed a linear regression analysis of the MJO-index and stratospheric zonal winds against the time-series of the amplitudes of the Hough modes. We confirm that the MJO is explained mainly by the first symmetric Rossby Mode (meridional index 1), Kelvin modes, in agreement with *Zagar & Franzke* (2015). The stratospheric zonal wind variability is explained mainly by the WIG modes and the first asymmetric Rossby modes (meridional index 2). We analyzed the interaction among those variables and tropical stratospheric ozone. The exchange of energy between the modes and their interaction with the ozone forcing explains the previous results. We highlight the strong influence of the ozone on the MJO-related modes on the intraseasonal time scale and on decadal time-scales, the last one being possibly a result of the solar cycle. We found influences of the gravity modes on the MJO-related modes to be the most relevant on bi-annual time-scales. This explains at least partially the work of *Yoo & Son* (2016) as well as subsequent articles on the QBO-MJO relation.

Composite analysis of the velocity and geopotential height of the Kelvin and Rossby modes associated with the MJO reveal how the differences in the characteristics of these modes during MJO events when the winds are positive in 30 Mb and when they are negative. For the Rossby modes differences (Fig. 12) are shown to be more significant during initial (1-3) and final (7-8) phases of a MJO cycle, and the spatial pattern is that expected of the rotational component of the MJO with double vortex pattern. The differences reveal a stronger rotational component of the MJO when the zonal winds at 30 Mb are positive. For Kelvin modes significant differences are found throughout the whole MJO cycle and the composite for the difference between the fields from both QBO phases follow a propagation pattern that seem to evolve eastward with similar speed of a typical MJO event ( 5m/s). This suggests that the QBO effect on the Kelvin mode is more uniform though-out a QBO cycle and in the Rossby modes this effect takes place in the initial and final phases of the MJO.

*Code availability.* TEXT

*Data availability.* TEXT

*Author contributions.* B.R. proposed the study, wrote the manuscript and did the statistical analysis, D.Y.T and L.M. worked on the PDC analysis, A.S.T. performed the normal mode decomposition analysis, P.L.S.D. helped with the discussion and the interpretation of the analysis.

*Competing interests.* The authors declare that they have no conflict of interest.

5 *Disclaimer.* TEXT

*Acknowledgements.* This work was financed by the FAPESP-PACMEDY project (grants 2015/50686-1, 2017/23417-5) and CAPES IAG/USP PROEX (grant 0531/2017) and the Meteorology Graduate Program at IAG-USP(CAPES Finance code 001).

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

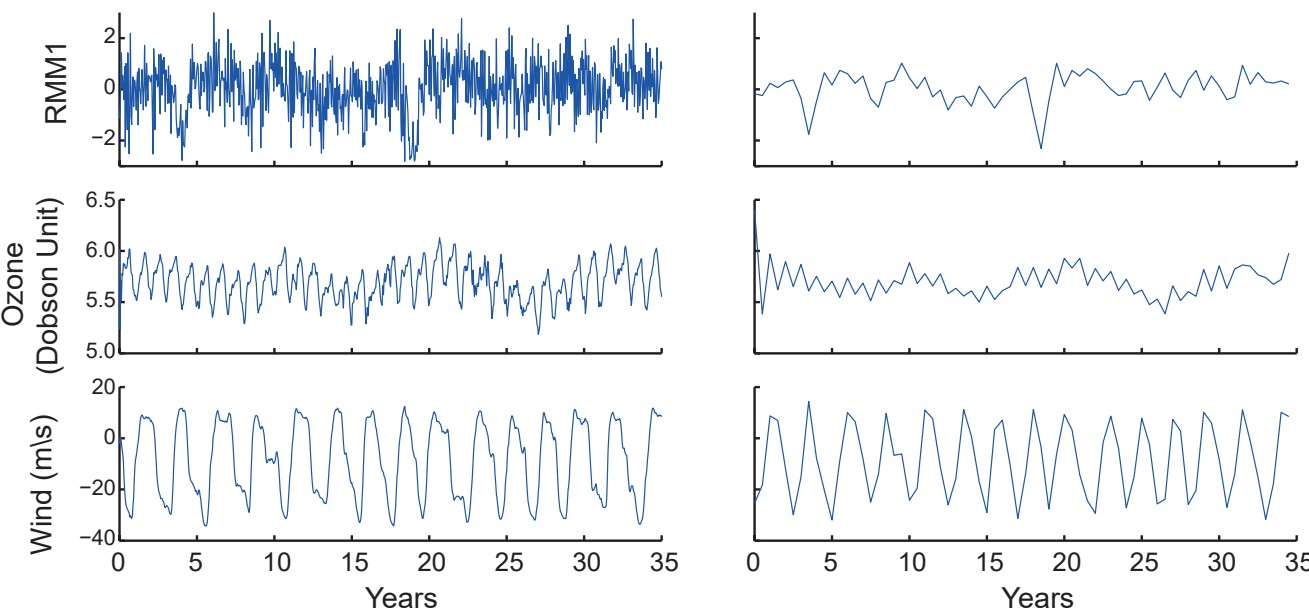

**Figure 1.** On the right, the time series resampled at a 10 days rate of the first component of the MJO index (top), ozone spacially averaged in the equatorial region (middle), and equatorial stratospheric wind (bottom). On the left, the same with band with resampling rate of 6 months.

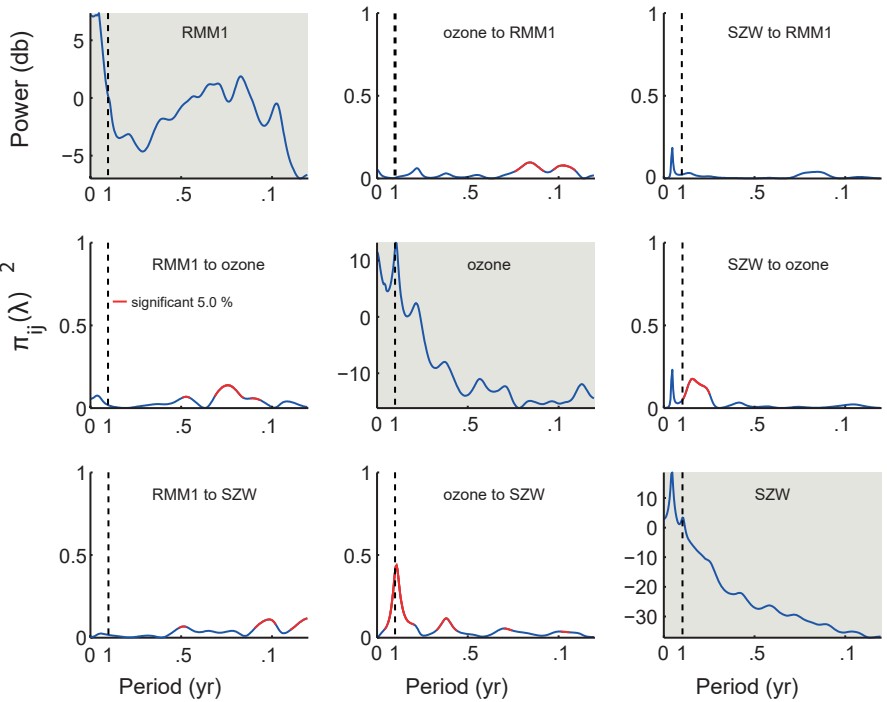

**Figure 2.** PDC between tropical stratospheric ozone and stratospheric zonal wind (SZW) and the RMM MJO index at the fast (>20days) timescale, frequencies are given are given in cycles/year. Panels in the main diagonal show the power spectral density of each time series. Off-diagonal panels indicate the PDC values between the time series. For each panel, the x-axis represents the frequency and y-axis the value of the PDC. Red lines represent the PDC values that were statistically significant.

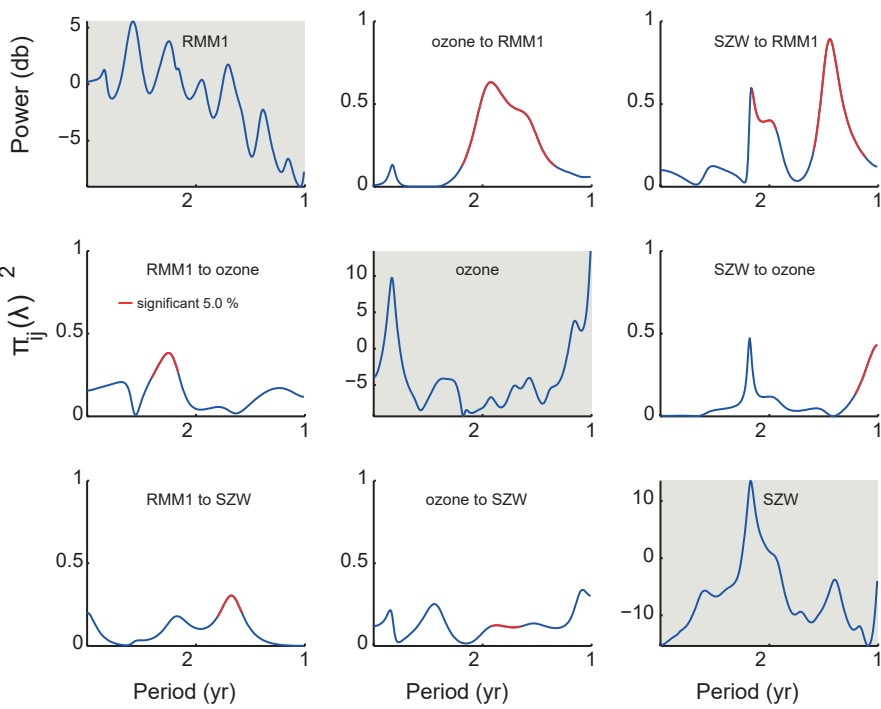

**Figure 3.** PDC analysis between MJO and Stratospheric zonal winds (SZW) at the slow (>1 year periods) timescale. Results indicate significant interaction on the annual-biennial timescales. Figure conventions are the same as in Figure 2.

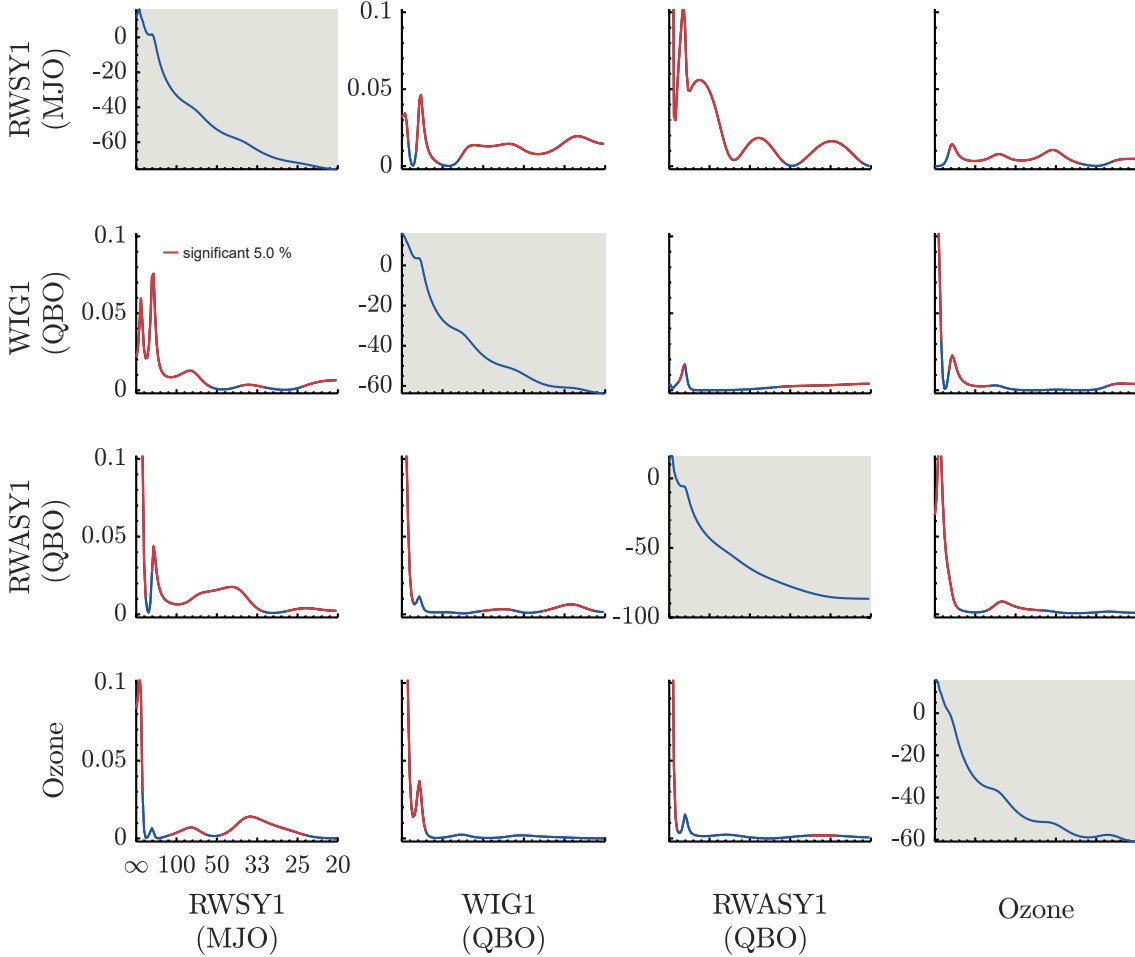

**Figure 4.** PDC analysis of the interaction of Kelvin, asymmetric Rossby, westward gravity modes and ozone at the fast timescale(periods given in days). Significant interactions (red curve) between MJO and ozone/QBO-related modes is found on intraseasonal, semi-annual and annual time-scales. Panels in the main diagonal show the power spectral density of each time series. Off-diagonal panels indicate the PDC values between the time series, where PDC direction is from the time series indicated in the column to the one indicated in the row. For each panel, the x-axis represents the frequency and y-axis the value of the PDC. Red lines represent the PDC values that were statistically significant.

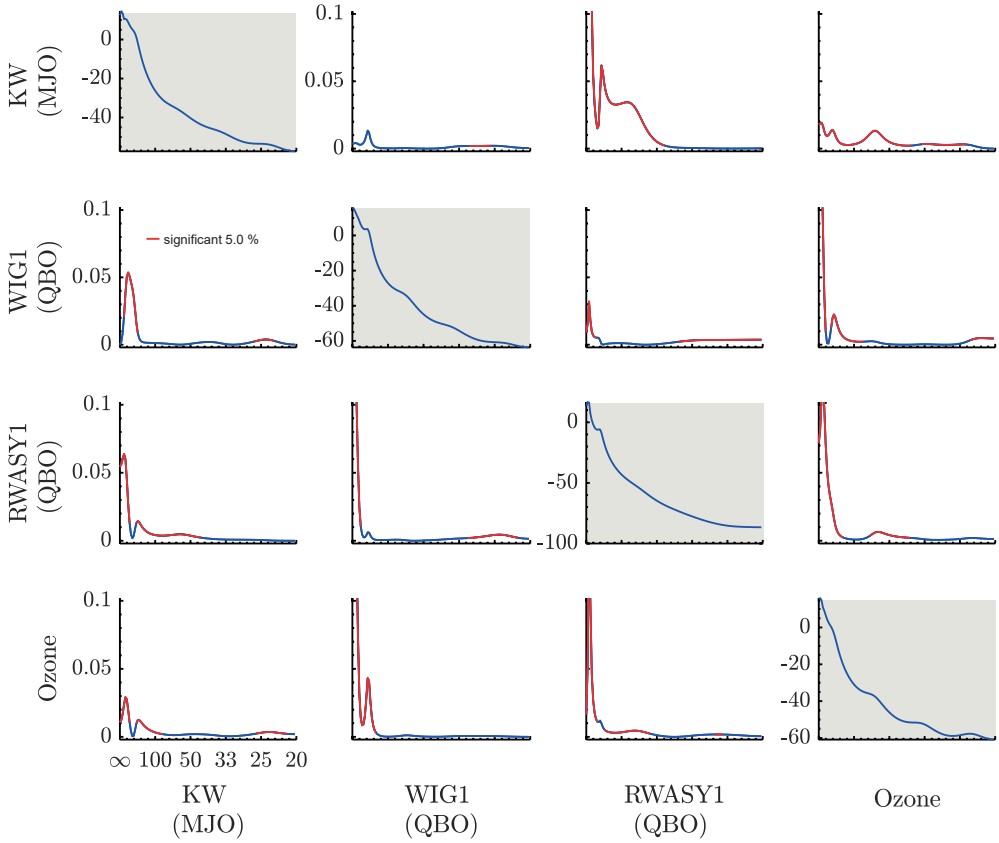

**Figure 5.** PDC analysis of the interaction of symmetric Rossby n=1, asymmetric Rossby n=1 and westward gravity modes and ozone at the fast timescale (periods given in days). Again, significant interactions (red curve) between MJO and ozone/QBO-related modes is found on intraseasonal, semi-annual and annual time-scales. Figure conventions are the same as in Figure 4.

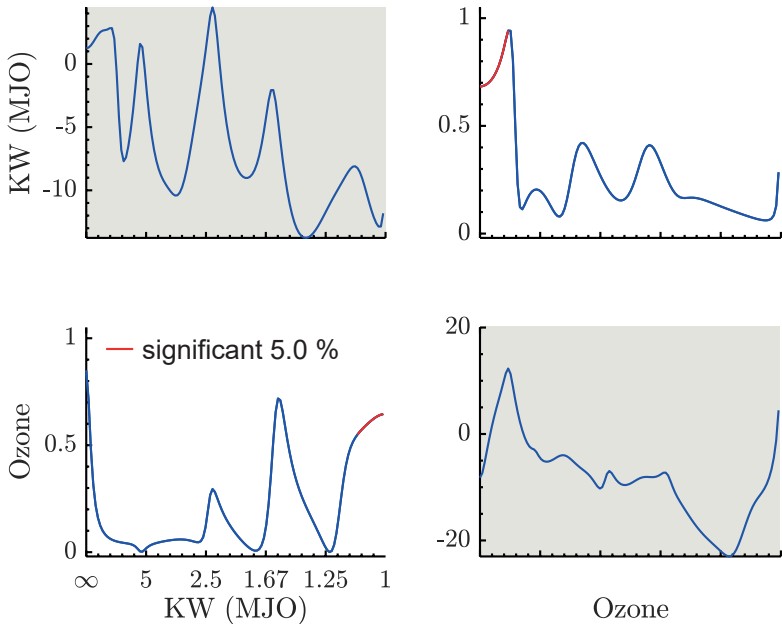

**Figure 6.** PDC analysis of the interaction of ozone modes and Kelvin waves (KW) at the slow timescale (periods given in years). The results show that KW influence the ozone on the annual time-scale, while the ozone influences KW on decadal time-scales. Figure conventions are the same as in Figure 4.

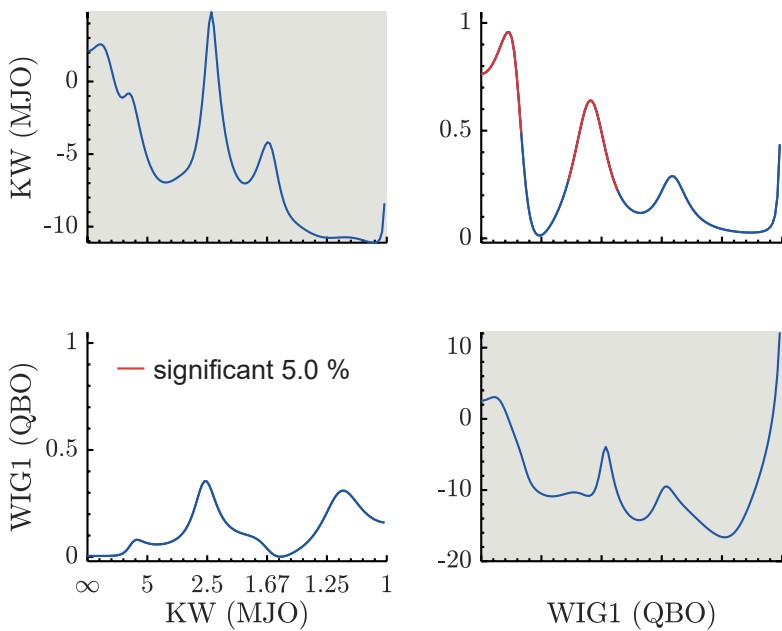

**Figure 7.** PDC analysis of the interaction of Kelvin modes (KW) and westward gravity modes (WIG) at the slow timescale (periods given in years). The results show a strong influence of the WIG mode on the KW on biennial and decadal timescales. Figure conventions are the same as in Figure 4.

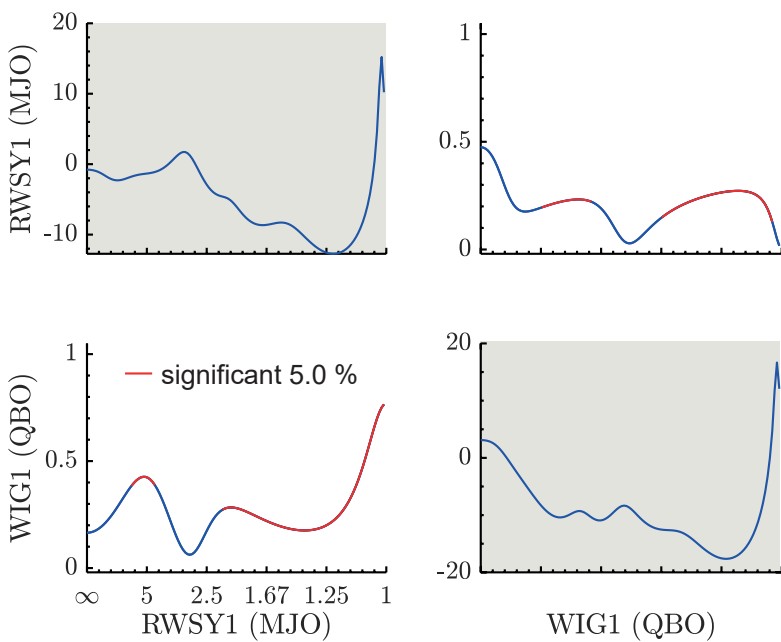

**Figure 8.** PDC analysis of the interaction of symmetric Rossby modes (meridional index 1, denoted by RWSY1) and westward gravity modes (WIG1) at the slow timescale (periods given in years). Important interactions are found in annual to interannual time-scales. Figure conventions are the same as in Figure 4.

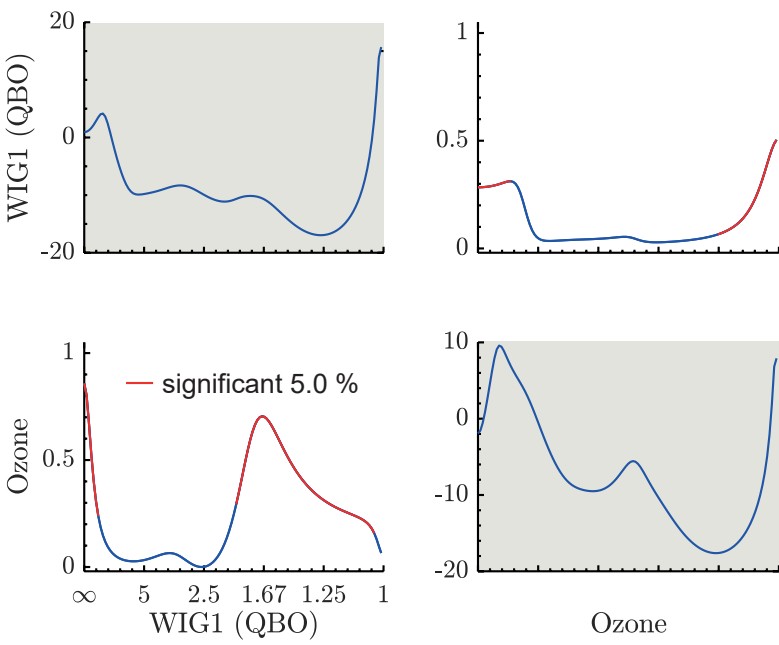

**Figure 9.** PDC analysis of the interaction of westward gravity modes and ozone at the slow timescale (periods given in years). Important interactions are found on annual-biennial time-scales as well as on the decadal time-scale. Figure conventions are the same as in Figure 4.

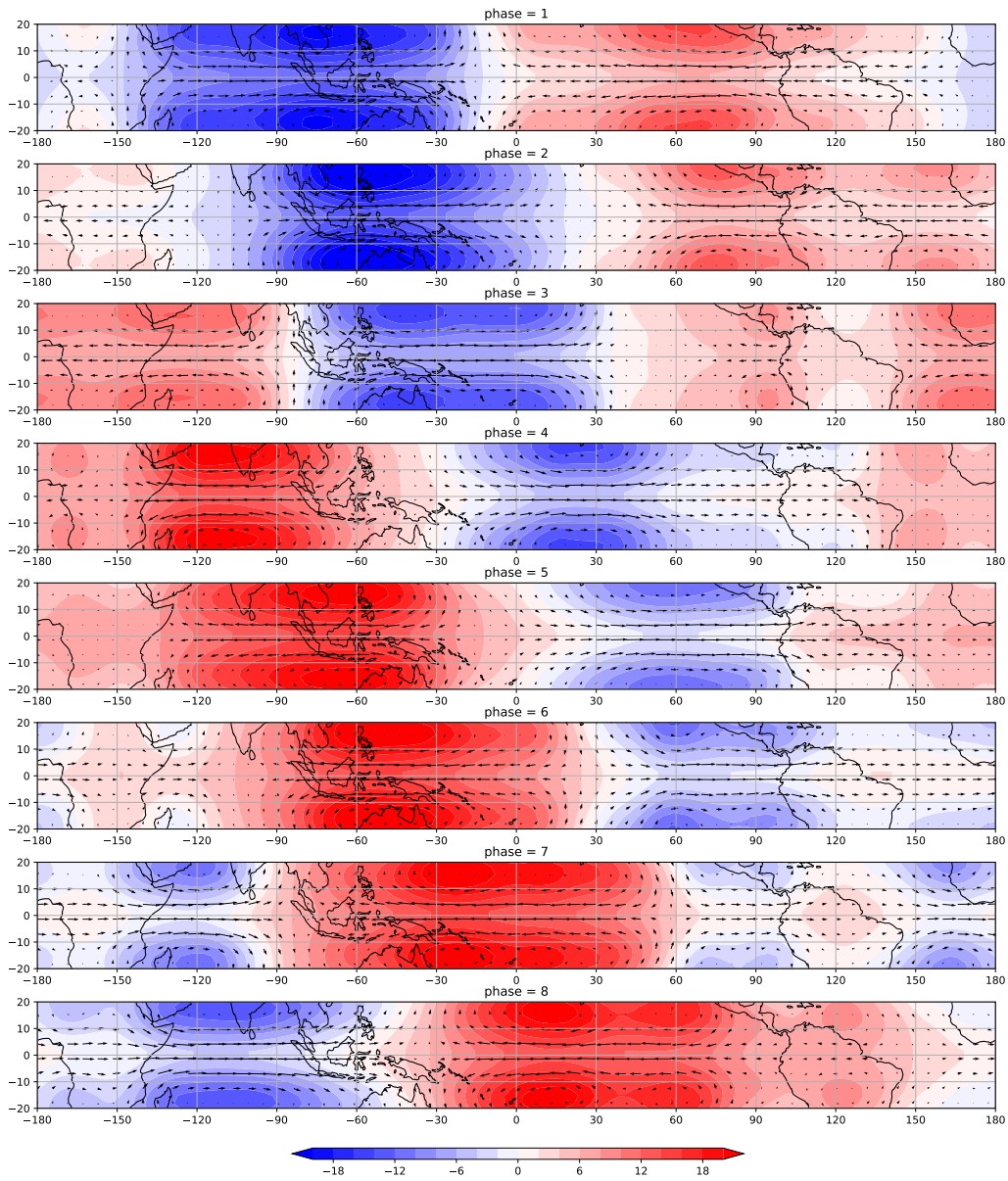

**Figure 10.** Reconstruction of the velocity and geopotential height fields associated with ROT modes with SZW30+ at 200 Mb.

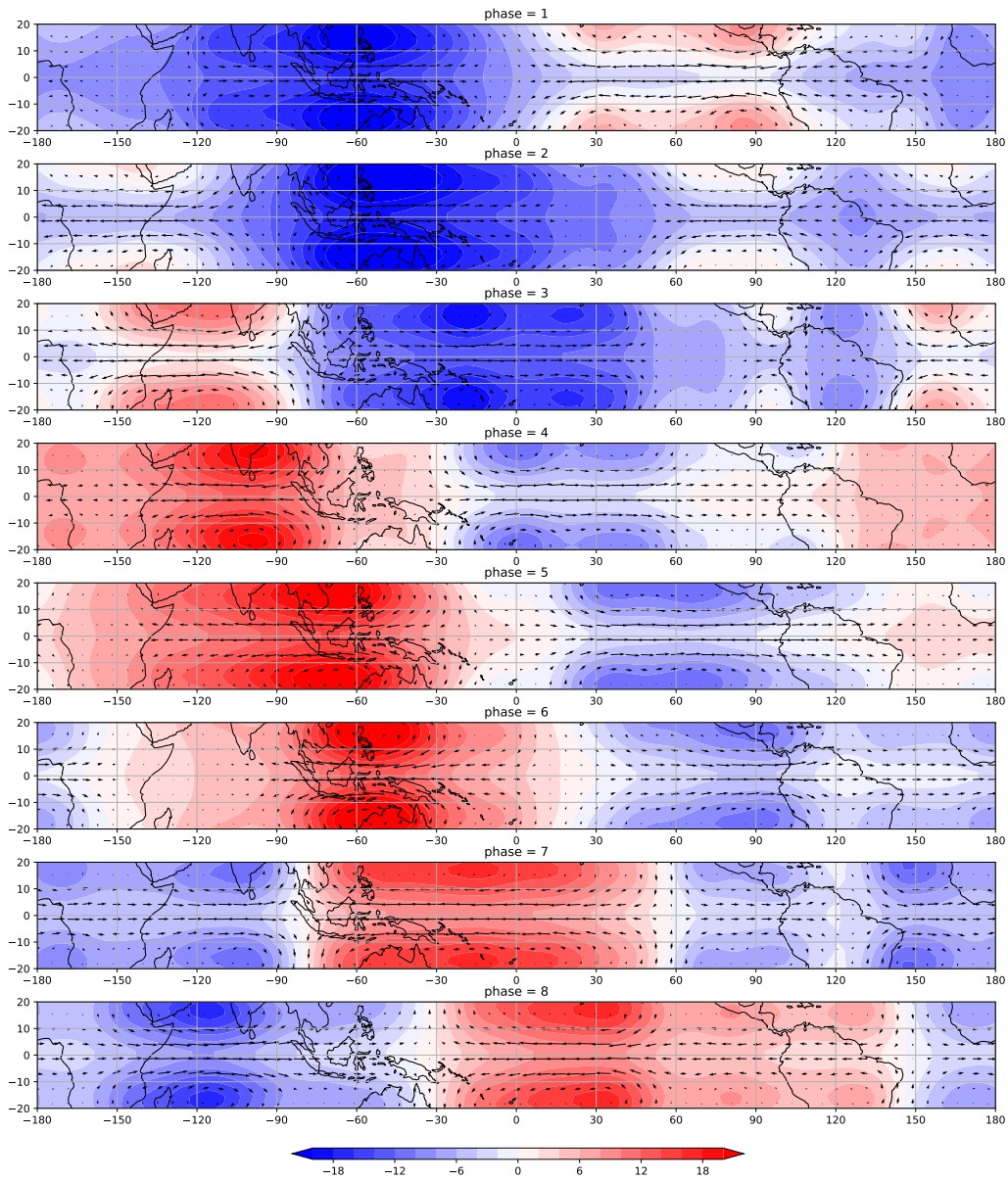

**Figure 11.** Reconstruction of the velocity and geopotential height fields associated with ROT modes with SZW30- at 200 Mb.

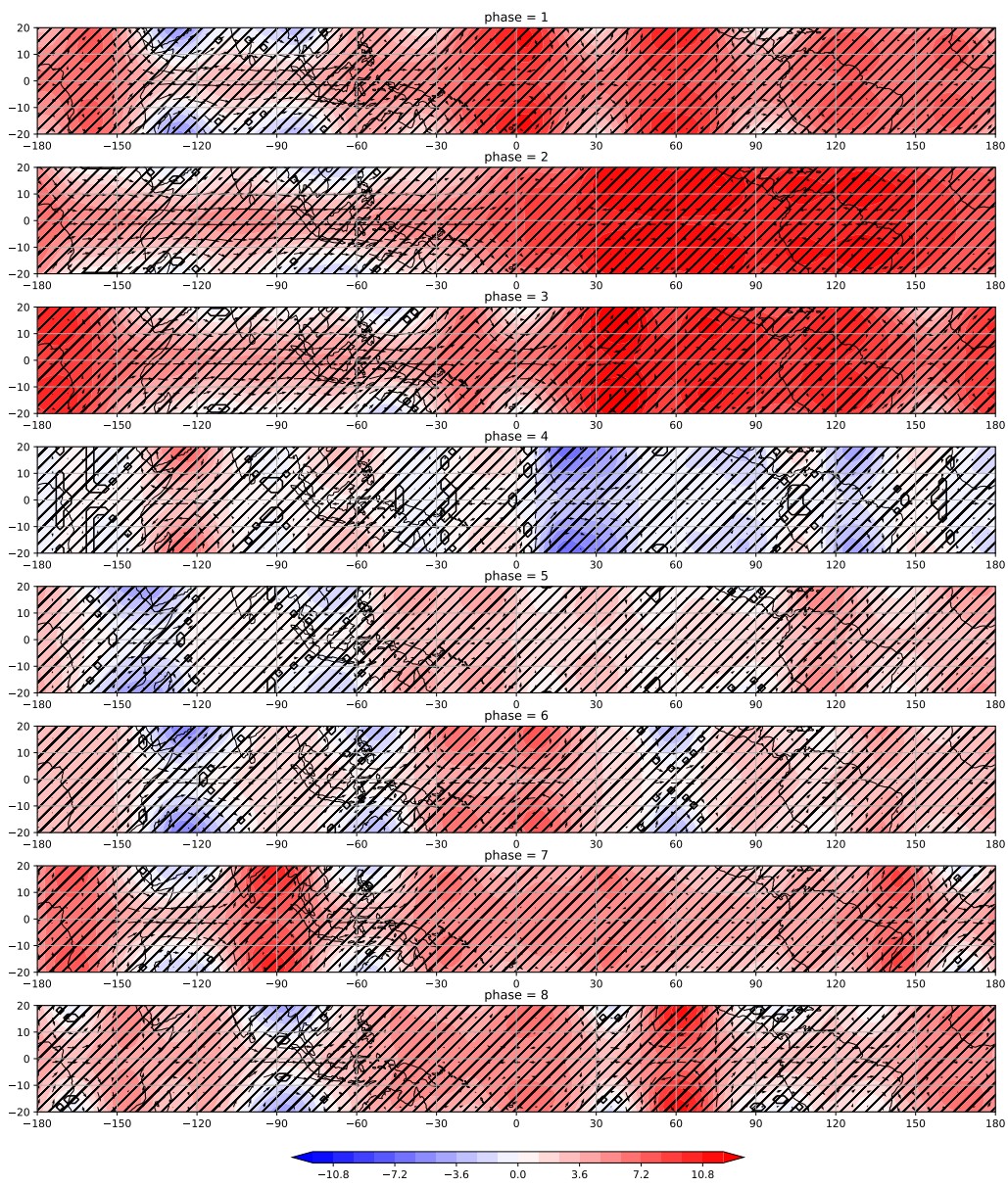

**Figure 12.** Figure 2.7.4: Difference between of the velocity and geopotential height fields associated with ROT modes with SZW30+ and SZW30-. The hatched region corresponds to significant difference of the geopotential height values under 5% confidence level.

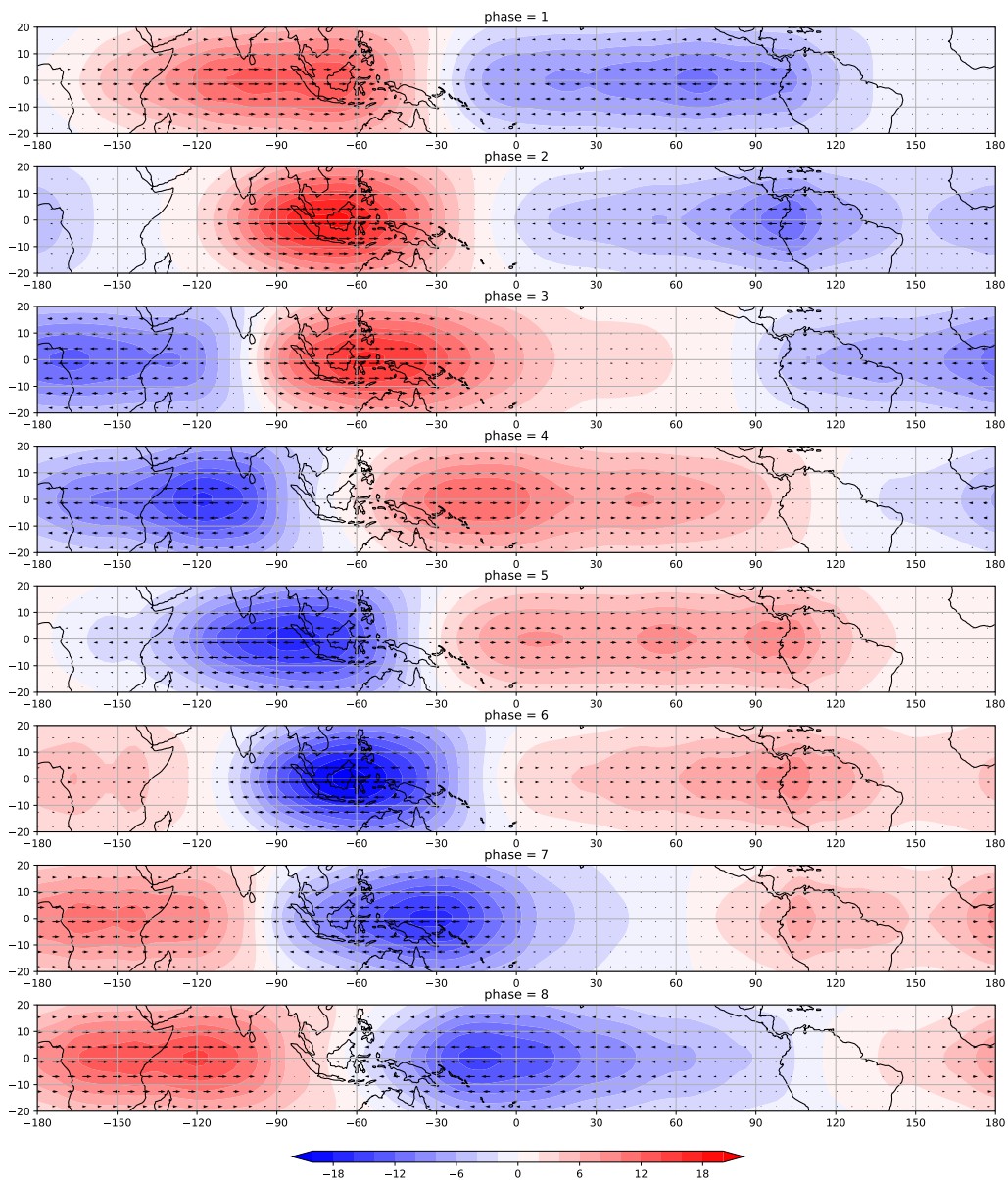

**Figure 13.** Reconstruction of the velocity and geopotential height fields associated with Kelvin modes with SZW30- at 200 Mb.

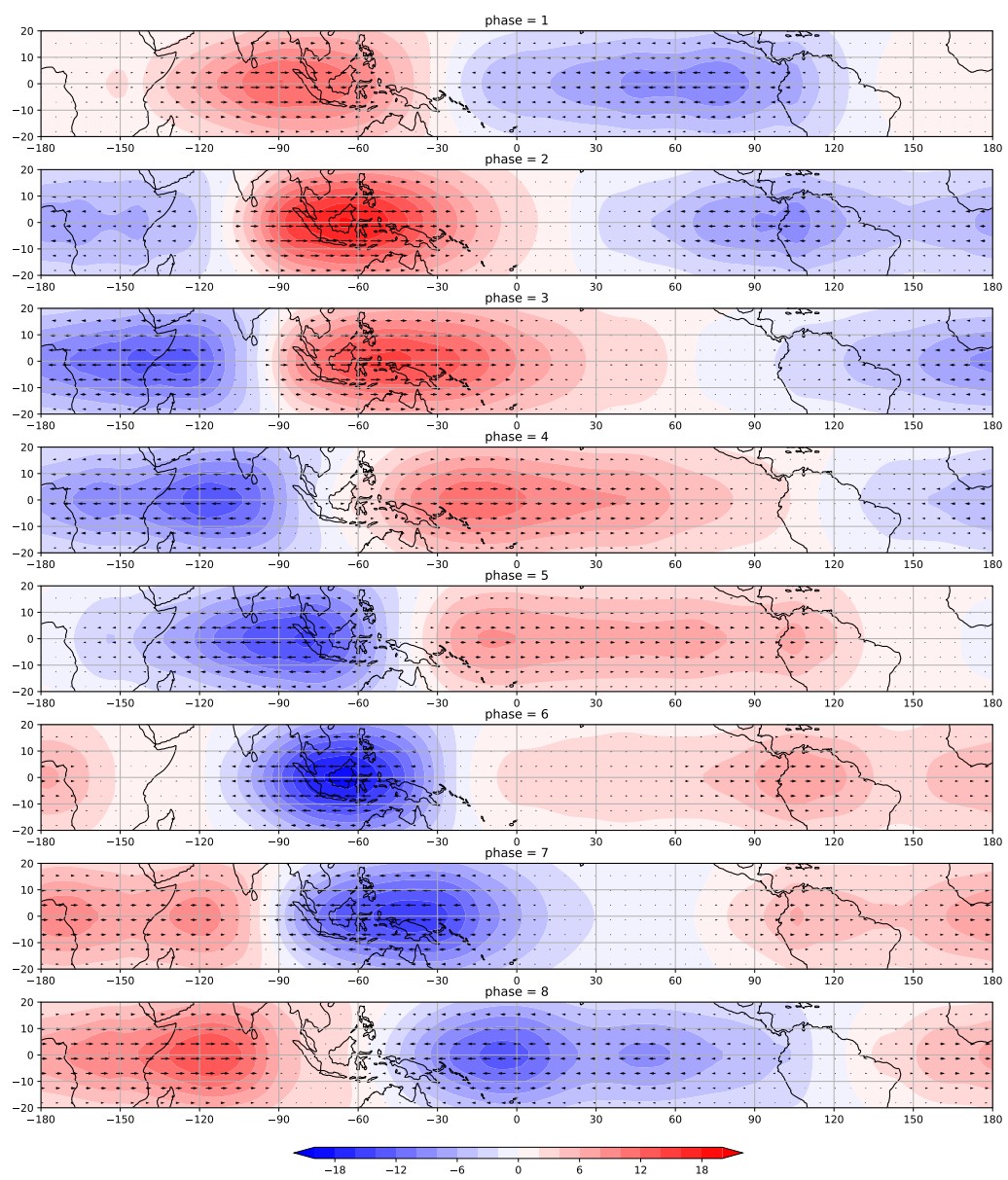

**Figure 14.** Reconstruction of the velocity and geopotential height fields associated with Kelvin modes with SZW30- at 200 Mb.

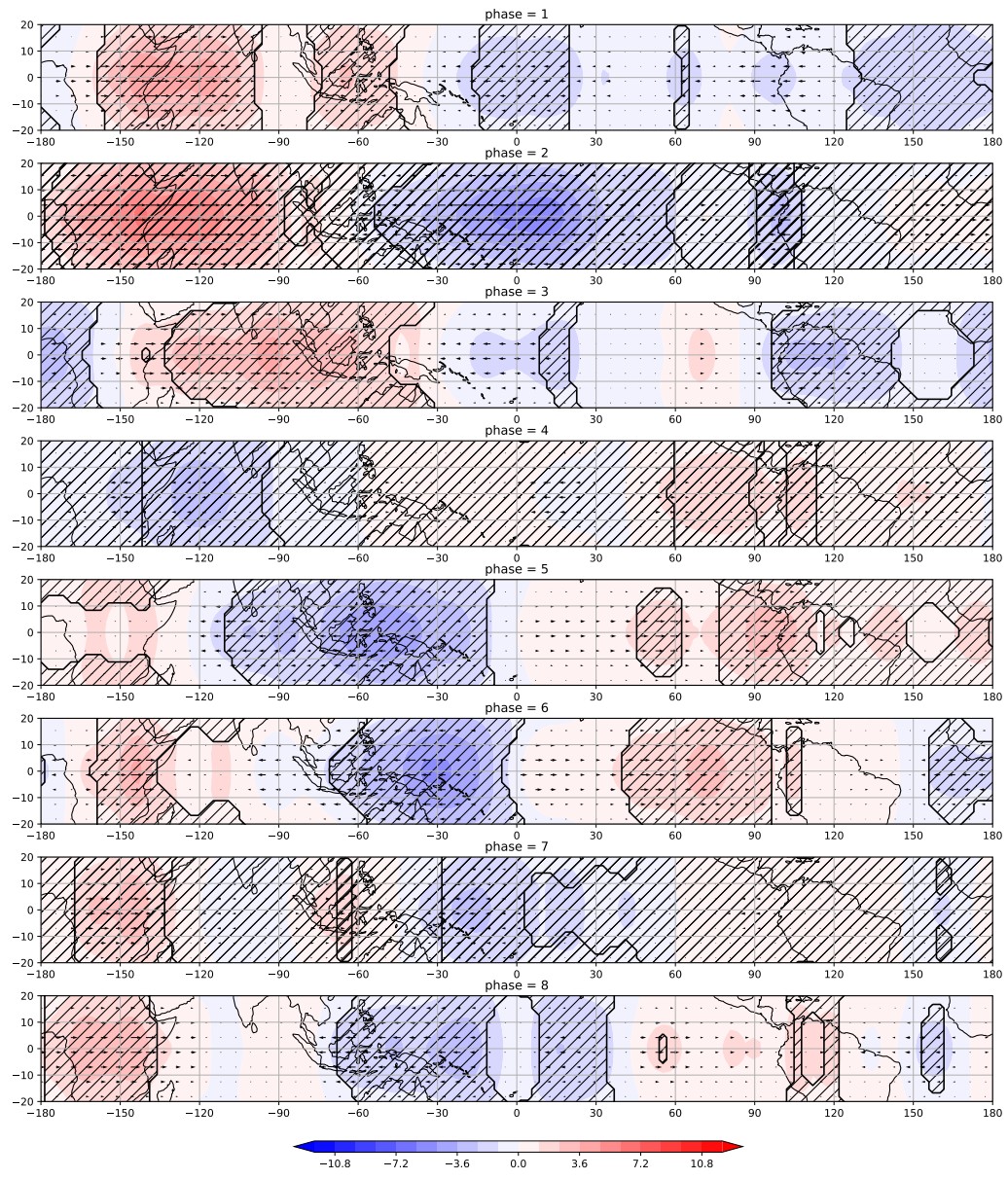

**Figure 15.** Difference between of the velocity and geopotential height fields associated with Kelvin modes with SZW30+ and SZW30-. The hatched region corresponds to significant difference of the geopotential height values under 5% confidence level.