# Peer review of "Stratospheric ozone and QBO interaction with the tropical troposphere on intraseasonal and interannual time-scales: a normal mode perspective"

_Earth System Dynamics, 2020_

## Short Comment (SC1) · 15 Jul 2020

"PDC was also used to detect the causality between the El Niño Southern Oscillation and the monsoons and also in the sea-air interaction in the South Atlantic Convergence Zone (Tribassi et. al, 2017)"

Since it has become obvious that common-mode tidal forcings control the majority of climate indices, as a first step one should consider how the tidal factors play into the models. See attached figures for AMO, ENSO, and QBO. Once this causality is understood, then it will be much easier to deal with other interactions. Cheers

[Figure]

[Figure]

[Figure]

**Fig. 1.** Common mode tidal forcing used to model AMO

[Figure]

**Fig. 2.** Common mode tidal forcing used to model ENSO

[Figure]

**Fig. 3.** Common mode tidal forcing used to model QBO

---

## Referee Comment (RC1) · Anonymous Referee #1 · 26 Jul 2020

Stratospheric ozone and QBO interaction with the tropical troposphere on intraseasonal and interanual time-scales: a wave interaction perspective, by Raphaldini et al

This study uses a "Partial Directed Coherence" method which is based on the concepts of Granger Causality to infer relationships between the QBO, ozone, the MJO and various wave modes identified by a normal mode decomposition. It is found that ozone and stratospheric winds influence the MJO, which confirms the results of previous studies. The potentially new insight that this study could offer is an insight into the mechanism behind the QBO-MJO connection through interactions involving inertio-gravity waves and Rossby waves. However, I have several major concerns that I think need to be

addressed before this study should be accepted for publication.

Major comments:

(1) I think the statistics of the method need to be much more carefully described. At the moment, we aren't really given any indication of how the significance is determined other than a reference to another article. One thing I am particularly concerned about is that by doing this frequency decomposition as well as using multiple variables, it means that effectively a very large number of tests has been performed. Is this accounted for when performing the significance tests. For example, if you test 100 different frequencies and use a 95% level, you'd expect 5 different frequencies to show a significant signal. Furthermore, how is autocorrelation in the time series for the low frequencies accounted for in the significance testing. For the decadal timescales there will be very few degrees of freedom in the observational record and I would hope that this is being accounted for in the statistical testing but it's not clear. So, I strongly recommend an improved discussion of the statistical testing and the significance of results in light of these complicating factors.

(2) I question whether showing the interaction between the gravity waves and the MJO is really an explanation. At pg 2, l3, it is stated that this connection represents a partial explanation, but it's not really a mechanistic understanding. It certainly hints at something that should be investigated, but I wouldn't even call it a partial explanation. One aspect I'm concerned about with this inference is whether the stratospheric zonal winds are accounted for when assessing the connection between the gravity waves and the MJO or not. It's not entirely clear to me. Is the connection between the gravity waves and the MJO just a simple assessment of the connection between the gravity waves and the MJO or is it an assessment of whether the gravity waves provide you more information beyond what you'd already get given the connection between the stratospheric zonal wind and the MJO. If it is not the latter, then isn't it possible that this connection between the gravity waves and the MJO simply represent the connection between the QBO and the MJO where the gravity wave variability is a signal of the

QBO and not necessarily connected to the MJO in a causal sense.

(3) Conclusions are drawn about what factors influence the MJO on what frequencies. I wonder if, having performed this causality analysis, which I expect will seem like a bit of a black box to many readers, whether the results could then be related back to something a bit more physical e.g., could you present the time series and lagged correlations between the fields at the relevant frequencies to convince readers of the actual correlation between these time series.

(4) I'm not entirely sure what is shown in Fig 12, but it looks kind of strange. It is described by "We recompose the zonal wind fields of WIG waves associated with the QBO". Is this showing where the amplitude of the gravity waves fluctuate along with the QBO? So it's really showing where orographically generated gravity waves are active? If so, it makes sense that there should be such a close correspondence between orography and this metric. But is it really the case that gravity waves over Greenland and Antarctica are varying with the QBO? Furthermore, I don't think it's really the orographic gravity waves that interact with the QBO, it's more the convectively generated gravity waves, which we don't really see in this figure. I think this all needs a bit more explanation and a bit more discussion of the physical linkages to complement the Partial Directed Coherence analysis.

Minor comments by line number.

pg 1, l7: This sentence is unclear. It's unclear whether ozone is influencing the MJO at periods of 1-2 months while the stratospheric winds are influencing the MJO at periods of 1.5-2.5 years or whether both are influencing the MJO at both frequencies. I think some re-wording is needed to make it clear which it is.

pg 1, l7: It's not clear to me how it can be determined that it's the stratospheric wind that is influencing the MJO and not e.g., stratospheric temperature anomalies that accompany the QBO wind variability. For example, Martin et al (2019), J. Atmos. Sci., 76, 669-688 show that the response to QBO zonal wind anomalies is much weaker

than the response to QBO temperature anomalies when imposing them in isolation in a cloud resolving single column model. So, I suggest taking care when making causal statements such as "stratospheric wind influencing the MJO".

pg 2, l20: A reference is missing here.

pg 3, l2: what does "innovations processes" mean?

pg 3, l4: I don't think it's enough to say that there is granger causality if that a_ij parameter is not zero. It has to be significantly different from zero above some threshold. Suggest making that clear. Also at equation (3).

pg 4: l29: It's not very clear what [-1/2,1/3) means here. Firstly, should it really be "[..)"? Secondly, it sounds like the freqency is between -1/2. and +1/2. but it's not clear if that's the case and if that is the case, what the meaning of these 1/2's are e.g., what are the units?

pg 6, l25: When examining the influence of stratospheric ozone, is there any accounting for the fact that stratospheric ozone and QBO zonal wind are not independent?

pg 9, l16: "boreal summer" –> "boreal winter"

Fig 1 caption: I think it should be >20 days and <180 days, not just > 20 days.

Typo's/wording:

Title: "interanual" –> "interannual" pg 1, l14: "influences" –> "influence" pg 1, l19: I'm not entirely sure what is meant by "is determinant for" but it sounds to me like a statement to the effect that the MJO determines the monsoons, which is not the case. Perhaps "is connected to tropical monsoons"? pg 2, l24: "how these" –> "how do these" pg 2, l34: "for all levels" –> "over all levels" pg 5, l9: "substitutes" –> "substitute" pg 5, l19: "specially" –> "especially" pg 7, l11: "what" –> "which" pg 7, l14: "what" –> "which" pg 8, l28: "be" –> "by" pg 10, l1: "contributions" –> "contributing" pg 10, l27: "borel" –> "boreal" Fig 8 caption: "influnce" –> "influence" Fig 11 caption: "times-cales"

–> "time-scales" or "timescales"

---

## Referee Comment (RC2) · Anonymous Referee #2 · 16 Aug 2020

Review of "Stratospheric ozone and QBO interaction with the tropical troposphere on intraseasonal and interanual time-scales: a wave interaction perspective" by Raphaldini et al.

Recommendation: major revisions

1) I am concerned about the use of the Granger causality method since this assumes linear dynamics and Gaussian statistics. The MJO is probably a non-linear phenomenon. Did you also test the convergent cross-mapping approach by Sugihara?

In a recently published studies we have shown that time-lagged CCM and machine learning approaches are much better: Huang, Y., C. Franzke, N. Yuan and Z. Fu,

2020: Systematic identification of causal relations in high-dimensional chaotic systems: Application to Stratopshere-Troposphere coupling. Clim. Dyn., in press. https://link.springer.com/article/10.1007/s00382-020-05394-0

Huang, Y., Z. Fu and C. Franzke, 2020: Detecting causality from time series in a machine learning framework. Chaos, 30, 063116

2) It is not really clear to me how you compute the time series you then use for the analysis. Are these just the projections of particular normal modes? If yes, how many normal modes do you use to represent the MJO and QBO? Or do you use just one normal mode for the respective wave type?

3) While the MJO normal modes have large amplitudes during MJO events and the set of normal modes are also then coherent. However, the normal modes can also have large amplitudes during non-MJO/QBO events. So, I think your results on the MJO time scale might be robust but I am not sure whether your results are related to the MJO on longer time scales; there probably is an effect of the QBO/ozone on the particular normal modes but I do not think you have shown that this is really related to the MJO.

4) The quality of some of the figures is rather poor (Figs. 3, 10, 11).

5) What do the diagonal plots in Fig. 1 represent? Is that the causality of the time series with itself? What can I learn from this?

6) How do you compute the significance of the causal relations? A brief description would be useful.

7) There is a recent paper: Franzke, C., D. Jelic, S. Lee and S. Feldstein, 2019: Systematic Decomposition of the MJO and its Northern Hemispheric Extra-Tropical Response into Rossby and Inertio-Gravity Components. Q. J. Roy. Meteorol. Soc., 145, 1147-1164.

They use a composite approach which might be better suited to investigate the MJO and QBO. Using linear regression might mix too many non-events into the analysis.

8) Please correct "Frankze" to "Franzke" in the references.

---

## Referee Comment (RC3) · Anonymous Referee #3 · 26 Aug 2020

This paper examines the influence of the tropical stratosphere (namely the QBO and tropical ozone) on one of the principal modes of tropical troposphere intra-seasonal variability – the Madden Julian Oscillation. This is pursued using causal inference techniques drawn from information theory and a normal mode decomposition of ERA-Interim wind and thermodynamical (geopotential heights) data. Various wave modes have been found linking these phenomena on intraseasonal and interannual timescales. The study concludes that the Himalayas are important in accounting for QBO-MJO connections on long timescales.

The current paper is generally well written, but falls well short in describing the analy-

sis in full and putting forward plausible physical mechanisms linking the normal modes with the tropical waves and the QBO and MJO across a wide range of timescales. Furthermore, important details have been omitted in most of the figures. In this reviewer's opinion the papers methodology does not quite match up with the stated aims for the paper. I recommend at least a major revision to the current paper.

**Main Points:**

- The analysis and interpretation of section 3 is suspect (and possibly in other sections). Figure 1 clearly shows regular and artificial peaks at regular (frequency) intervals most likely resulting from the bandpass pre-processing of the data. The features look similar to those which would appear in data convolved with a square filter. I recommend that suitable prefiltering is done to minimise these numerical artefacts (i.e. using appropriate tapering methods).

- It seems to this reviewer that the annual cycle has been retained in the data. Presumably retention of the annual cycle and sub-harmonics will obscure attribution of causality between the various timeseries? Why has the annual cycle been retained and what impact will this have on the interpretation of the results?

- The authors should provide figures for the timeseries used in the paper, before and after processing, including those short and long timeseries used throughout the manuscript.

- The authors have not justified the use of indices thought relevant for MJO-QBO connections, namely MJO indices and the westward propagating gravity wave modes (and various others wave modes). There are a number of competing mechanisms for explaining the observed correlations between the MJO and QBO. A number of these do not explicitly involve waves, but rather upper tropospheric temperature, wind-shear or static-stability. The title of the paper suggests a focus

on waves, but this needs to naturally come following an appraisal of the various mechanisms first.

- The various horizontal/vertical normal modes used to construct QBO and MJO patterns and timeseries need to be captured somewhere (e.g. supplementary materials) as they feature prominently in the analysis.

- There is a lot of various missing information on the figures (labels, units, tickmarks etc), which has mostly been identified in the points below. All figures need to be improved for future review.

- The spectra look very smooth; has any smoothing been applied to the power spectra? If so, how has this been achieved?

- Figures 8-11. What physical mechanism will causally link wave modes on inter-annual to decadal timescales? What hypothesis is being tested?

- Can the authors put forward a plausible physical mechanism linking the Hi-malayas near 30-40N and two equatorially confined phenomena – MJO and QBO? Furthermore, how should this mediate the observed statistical relationship between the QBO and MJO?

- The authors have looked at large scale circulation processes in assessing long-time scale relationships between the QBO and MJO. What though are the roles for small-scale gravity waves in linking QBO and MJO connections?

**Other Points:**

(title) *interannual*

(P1, L2) . . .*provides evidence*. . .

(P1, L4) . . .*global-scale*. . .

(P1, L5) …Partial Directed Coherence method *was used* and enabled us…

(P1, L8) what does *the latter* refer to; the wind? If so, please make this explicit.

(P1, L18) Capitalise *Quasi-Biennial Oscillation*

(P1, L19-20) replace "…determinant for the tropical monsoons and with global impacts." With "…impacts the tropical monsoons and so has global influence."

(P2, L1) replace *propagation* with *descending*

(P2, L1-2) Need a reference for influence of QBO on climate e.g. (Holton and Tan, 1980)

(P2, L2-3) suggest removing "*and inference of the dominant frequencies*" as it does not add to the sentence.

(P2, L6) replace *stratospheric* with *solar*. Furthermore, this paragraph ought to more explicitly delineate/separate solar influence and stratospheric influence, on the troposphere

(P2, L11) missing space (…*instance, Baldwin*…)

(P2, L14) replace *depends on* with *is sensitive to*.

(P2, L16) replace *depending on* with *to*

(P2, L20) missing reference at the end of the sentence. A good reference might be (Kim et al., 2020)

(P2, L21) remove *series*

(P2, L23) replace *circuntances* (sic) with *conditions*

(P2, L30) what is meant by *unfiltered* in this context?

(P2, L31) *European Centre for Medium-Range Weather Forecasts*

(P3, L4) I do not understand how multiplying the spectrum by its complex conjugate (i.e. to find the power-spectrum) introduces spurious power at particular frequencies. Is there a reference where this effect is further explained?

(P3, L7-10) Was the seasonal cycle removed from the data?

(P3, L14) remove *for*

(P3, L18) suggest removing *responsible for* with *contributing to*.

(P4, L2) *has* -> *have*

(P4, L18) Clarify: *both former methods*: Grainger Causality and what other method?

(P5, L17) . . .based *on* the concept. . .

(P5, L20) *captures*

(P5, L24) *nonlinear*

(P4, L29) The definition and use of the frequency range [-.5,.5) needs to be explained here. Why include -0.5 but exclude 0.5?

(P6, L18) The first sentence is not a sentence; please rephrase. For the second sentence I suggest changing *well-adjusted* to *well-represented*.

(P7, figure 2 3) Half of the panels do not have axis labelling or tickmarks (none indicate y-axis units).

(P7, L11) . . .*which* can be seen. . .

(P7, L14) . . .*which* is a strong indication. . .

(P7, L13) QBO has a mean period of 28 months, so it is difficult to link it with ozone and the MJO indices in figure 2. Furthermore, interpreting statistically significant features appearing at different frequencies within the 2 MJO indices challenges the meaningfulness of the results.

(P7, L15) Any link with the global monsoon system cannot be substantiated within figure 2.

(P7, figure 3) figure 3 looks lossy (fuzzy/bitmap), perhaps a vectorised figure can be made.

(P7, figure 4) Please state the scale (and units) of the wind vectors and the pressure units?

(P7, figure 5) To better show the patterns associated with the QBO can the authors redo figure 5 with a log-pressure vertical scale (i.e. z).

(P8, L1) *We seek interactions between...*

(P8, figure 6  7) Units? Labels? Tick-marks?

(figure 8 caption) *influence*

**References:**

Holton, J. R. and Tan, H.-C. (1980) 'The Influence of the Equatorial Quasi-Biennial Oscillation on the Global Circulation at 50 mb', Journal of the Atmospheric Sciences, 37(10), pp. 2200–2208. doi: 10.1175/1520-0469(1980)037<2200:TIOTEQ>2.0.CO;2.

Kim, H. et al. (2020) 'The Lack of QBO–MJO Connection in CMIP6 Models', Geophysical Research Letters, 47(11). doi: 10.1029/2020GL087295.

---

## Author Comment (AC1) · 24 Sep 2020

Dear Dr. Paul Pukite, we thank you for your interest in our work. We acknowledge that common mode forcing of tidal nature plays a major factor in the dynamics of waves. Here we have included the analysis of ozone for this reason, since we believe that this is an important forcing term for stratospheric disturbances that acted on the timescales relevant for our analysis. Tidal forcing of gravitational nature plays a major role in forcing gravity waves at the diurnal-hourly timescales, however we do not think they are relevant to the analysis presented here. For a recent reference that explores this issues please check (Sakazaki, T., & Hamilton, K. (2020). An Array of Ringing Global

[Figure]

Free Modes Discovered in Tropical Surface Pressure Data. Journal of the Atmospheric Sciences, 77(7), 2519-2539.). You have also to keep in mind that we are only considering the causality between processes at the same time scale, if we were considering inter scales connection, this probably would be and important factor, see (Raupp, C. F., & Silva Dias, P. L. (2009). Resonant wave interactions in the presence of a diurnally varying heat source. Journal of the atmospheric sciences, 66(10), 3165-3183.) for the role of the diurnal cycle on the dynamics of the MJO.

———————————————————

---

## Author Comment (AC2) · 24 Sep 2020

(1)I think the statistics of the method need to be much more carefully described. At the moment, we aren't really given any indication of how the significance is determined other than a reference to another article. One thing I am particularly concerned about is that by doing this frequency decomposition as well as using multiple variables, it means that effectively a very large number of tests has been performed. Is this accounted for when performing the significance tests. For example, if you test 100 different frequencies and use a 95% level, you'd expect 5 different frequencies to show a significant signal. Furthermore, how is autocorrelation in the time series for the low frequencies

accounted for in the significance testing. For the decadal timescales there will be very few degrees of freedom in the observational record and I would hope that this is being accounted for in the statistical testing but it's not clear. So, I strongly recommend an improved discussion of the statistical testing and the significance of results in light of these complicating factors.

R: We apologize that we did not describe the statistics with sufficient details. PDC is a function of the coefficients of vector autoregressive model. Given that the coeffcients are asymptotically jointly normally distributed, we can use the delta method (Serfling, 1980) to analytically calculate the asymptotic statistics for PDC. After a straighforward but tedious algebraic computation, we can show that PDC at frequency lambda is distributed asymptotically (under the null hypothesis of zero PDC) as the weighted sum of two chi-square with one degree of freedom (Takahashi et al., 2007). Therefore, we can use this asymptotic distribution to calculate the p-values. For details of the derivation, we refer to Takahashi et al. (2007). Significance levels for frequency domain quantities are controlled only point-wise as this is the standard everywhere. The reason for this is that the point estimates for neighboring frequencies are highly correlated. Therefore, standard correction like bonferroni or even FDR that assume independence or weak dependence give the wrong significance level. Every single article that we found where PDC, coherence or bi-coherence were used and the significance level is reported use the frequency-wise significance level (for representative examples see Huybers and Curry, 2006 and Came et al., 2007). For PDC it is easy to see that the use of frequency-wise significance level is reasonable given that the PDC values for different frequencies are the Fourier transform of the same coefficients of the autoregressive process. The fact that lower frequency have fewer samples are taken care by higher threshold values for PDC at lower frequencies. We added the following brief description of the statistics for PDC in the main text. "PDC is a function of the coefficients of vector autoregressive model. Given that the coefficients are asymptotically jointly normally distributed, we can use the delta method (Serfling, 1980) to obtain analytically the asymptotic statistics for PDC. After an algebraic computation we can show

that PDC at frequency lambda is distributed asymptotically (under the null hypothesis of zero PDC) as the weighted sum of two chi-square with one degree of freedom (Takahashi et al., 2007). Therefore, we can use the asymptotic distribution to calculate the p-value. For details of the derivation, we refer to Takahashi et al. (2007). The significance level used in the article for PDC is the frequency-wise value as it is the standard for frequency domain analysis given the high correlation between the point estimates for neighboring frequencies (see e.g. Huybers and Curry, 2006; Came et al., 2007)."

(2) I question whether showing the interaction between the gravity waves and the MJO is really an explanation. At pg 2, l3, it is stated that this connection represents a partial explanation, but it's not really a mechanistic understanding. It certainly hints at something that should be investigated, but I wouldn't even call it a partial explanation. One aspect I'm concerned about with this inference is whether the stratospheric zonal winds are accounted for when assessing the connection between the gravity waves and the MJO or not. It's not entirely clear to me. Is the connection between the gravity waves and the MJO just a simple assessment of the connection between the gravity waves and the MJO or is it an assessment of whether the gravity waves provide you more information beyond what you'd already get given the connection between the stratospheric zonal wind and the MJO. If it is not the latter, then isn't it possible that this connection between the gravity waves and the MJO simply represent the connection between the QBO and the MJO where the gravity wave variability is a signal of the QBO and not necessarily connected to the MJO in a causal sense.

R:The idea to investigate the effect of QBO related normal modes with MJO related normal modes was inspired by the works on nonlinear resonance as a driver for MJO through the interaction of tropics-extra tropics,see : Raupp, C. F., & Dias, P. L. S. (2010). Interaction of equatorial waves through resonance with the diurnal cycle of tropical heating. Tellus A: Dynamic Meteorology and Oceanography, 62(5), 706-718/ -Majda, A. J., & Biello, J. A. (2003). The nonlinear interaction of barotropic and equatorial baroclinic Rossby waves. Journal of the atmospheric sciences, 60(15), 1809-

1821. ). The idea then is to search for evidence for mode interaction that may lead to stratosphere-troposphere interaction similar to the aforementioned theories for the interaction tropics-extratropics. In this sense our work may be regarded and a evidence for such a mechanism, although we do not develop the theory itself. Regarding the information of the interaction of gravity waves on MJO. The normal modes that contribute to the QBO are determined by a linear regression procedure, gravity waves being some of the main contributors. To say that gravity waves associated with the QBO also interact with the QBO gives more information on the MJO-QBO interaction since it restricts the type of mode responsible for the interaction, in this particular case gravity modes rather than balanced (Rossby) modes.

3) Conclusions are drawn about what factors influence the MJO on what frequencies. I wonder if, having performed this causality analysis, which I expect will seem like a bit of a black box to many readers, whether the results could then be related back to something a bit more physical e.g., could you present the time series and lagged correlations between the fields at the relevant frequencies to convince readers of the actual correlation between these time series.

R:In the present version of the manuscript we have included a composite analysis based on Reviwer #2 suggestion showing the differences on each normal mode component of the the MJO depending on the phase of the QBO.

4) I'm not entirely sure what is shown in Fig 12, but it looks kind of strange. It is described by "We recompose the zonal wind fields of WIG waves associated with the QBO". Is this showing where the amplitude of the gravity waves fluctuate along with the QBO? So it's really showing where orographically generated gravity waves are active? If so, it makes sense that there should be such a close correspondence between orography and this metric. But is it really the case that gravity waves over Greenland and Antarctica are varying with the QBO? Furthermore, I don't think it's really the orographic gravity waves that interact with the QBO, it's more the convectively generated gravity waves, which we don't really see in this figure. I think this all needs a bit more

explanation and a bit more discussion of the physical linkages to complement the Partial Directed Coherence analysis.

R: After discussion with the co-authors we decided to remove this section on the spatial structure of the gravity waves, since we came to the conclusion that it was not bringing insight into the main problem of the article. Instead we followed the Reviewer #2 suggestion to present composites of the MJO related normal modes for each MJO phase, comparing them as a function of the phase of the QBO (positive or negative).

---

## Author Comment (AC3) · 24 Sep 2020

Due to the large number of figures and formulas the complete answer is found in the attached PDF file.

1)I am concerned about the use of the Granger causality method since this assumes linear dynamics and Gaussian statistics. The MJO is probably a non-linear phenomenon. Did you also test the convergent cross-mapping approach by Sugihara? In a recently published studies we have shown that time-lagged CCM and machine learning approaches are much better: Huang, Y., C. Franzke, N. Yuan and Z. Fu, C1 ESDD Interactive comment Printer-friendly version Discussion pa-
per 2020: Systematic identification of causal relations in high-dimensional chaotic systems: Application to Stratopshere-Troposphere coupling. Clim. Dyn., in press. https://link.springer.com/article/10.1007/s00382-020-05394-0Huang, Y., Z. Fu and C. Franzke, 2020: Detecting causality from time series in a machine learning framework. Chaos, 30, 063116.

R: In this article, we have used the PDC method to infer Granger causality between multiple time-series in the frequency domain. The main advantage of PDC and Granger causality is that it is theoretically related to the mutual information rate (MIR) between signals (see Takahashi et. al 2010 Information theoretic interpretation of frequency domain connectivity measures. Biological Cybernetics, v.103, p. 463-469, 2010.; Geweke, J. F. (1984). Measures of conditional linear dependence and feedback between time series. Journal of the American Statistical Association, 79(388), 907-915.). Information-theoretic quantities are usually costly to estimate directly from time-series since it relies on the estimation of multi-dimensional probability distributions. As proved in Takahashi et. al 2010, PDC is a Gaussian approximation to the MIR. This means that if the time-series are stationary and Gaussian PDC provides an exact estimate for the MIR, when the time-series are not Gaussian (possibly due to underlying nonlinearities) the PDC will capture part but not all of the information flow between the time-series. There are many "causality" estimation methods in the literature, all of them with some advantages and drawbacks. Among the several causality detection methods the Convergent-Cross Mapping (CCM) method is proposed as a method that is capable to capture couplings in highly-nonlinear settings since it relies phase-space embedding procedures. CCM. However, it comes with a few drawbacks that would require more in-depth investigation before we could apply it in the present setting, namely: (1) CCM is a bi-variate measure. Granger causality and PDC are genuinely multivariate measures. (2) CCM may lead to wrong or misleading results when moderate to high levels of noise are present (see Mønster, D., Fusaroli, R., Tylén, K., Roepstorff, A., & Sherson, J. F. (2017). Causal inference from noisy time-series data—Testing the Convergent Cross-Mapping algorithm in the presence of noise and external influence. Future Generation

Computer Systems, 73, 52-62.). Granger causality and PDC are designed to work for signals with stochasticity. (3) CCM does not have an automated way to decide the optimal lag between time series. Granger causality and PDC are based on autoregessive process in which order estimation is well studied. (4) There are no theoretical guarantees for the statistical properties of CCM. Both PDC and Granger causality are at very well studied measures in which there are thousands of articles applying it and we understand well their statistical properties (Lutkepohl, 2005; Takahashi et al., 2007).

Finally, although PDC is a stochastic linear method, it correctly reconstruct the topology of networks of nonlinear oscillators (see Winterhalder, M., Schelter, B., & Timmer, J. (2007). Detecting coupling directions in multivariate oscillatory systems. International Journal of Bifurcation and Chaos, 17(10), 3735-3739.), Moreover, it has been successfully and extensively used to infer information flow in highly nonlinear time-series data in neuroscience (Bressler, S. L., & Seth, A. K. (2011). Wiener–Granger causality: a well established methodology. Neuroimage, 58(2), 323-329.). The fact that PDC can detect nonlinear interactions is not difficult to understand, given that linear regression also can see nonlinear interaction unless the nonlinearity is highly non-monotonic.

2) It is not really clear to me how you compute the time series you then use for the analysis. Are these just the projections of particular normal modes? If yes, how many normal modes do you use to represent the MJO and QBO? Or do you use just one normal mode for the respective wave type? R:The time-series associated with the normal modes that we used correspond the the energy of a group of modes defined by:

$E(t) = 1/2 \sum_{m=1}^{M} g D_m \sum_{k=0}^{K} \sum_{n=0}^{N} ([\chi_{kmn}(t)]^{*} \chi_{kmn}(t))$
Where g is the acceleration of gravity, $D_m$ is the equivalent height of the m-th vertical index, $\chi_{kmn}(t)$ is the complex amplitude of the normal mode with zonal wave number k, meridional index n and vertical index m. M=43, K=32 and N are the respective truncation numbers for each index. For the MJO we selected the three first three even meridional indices for the Rossby modes (no selection on the vertical and zonal

modes).

3) While the MJO normal modes have large amplitudes during MJO events and the set of normal modes are also then coherent. However, the normal modes can also have large amplitudes during non-MJO/QBO events. So, I think your results on the MJO time scale might be robust but I am not sure whether your results are related to the MJO on longer time scales; there probably is an effect of the QBO/ozone on the particular normal modes but I do not think you have shown that this is really related to the MJO. In the present version of the manuscript we have included the composite analysis as suggested by this referee. This analysis clearly shows a difference in the long term behavior of the MJO-related modes, this was done for the QBO timescale (∼28 months), and probably accounts for the causality between QBO modes and MJO modes at this time scale. Differences at other time-scales such as the solar cycle timescale still need to the investigated in more detail. 4) The quality of some of the figures is rather poor (Figs. 3, 10, 11). R:Due to the large number of figures we were having problems compiling the file, which lead us to include figures with lower resolution, in the new version of the manuscript we included figures with better resolution. 5) What do the diagonal plots in Fig. 1 represent? Is that the causality of the time series with itself? What can I learn from this? R:The diagonal plots correspond to the power spectrum of each of the variables, which is equivalent to the PDC of between the variable and itself. 6) How do you compute the significance of the causal relations? A brief description would be useful. R:In this version of the manuscript we have included a description of the statistics, in particular how we obtain the confidence intervals of the PDC. We refer back to our response to the first question of referee #1. "We apologize that we did not describe the statistics with sufficient details. PDC is a function of the coefficients of vector autoregressive model. Given that the coeffcients are asymptotically jointly normally distributed, we can use the delta method (Serfling, 1980) to analytically calculate the asymptotic statistics for PDC. After a straighforward but tedious algebraic computation, we can show that PDC at frequency lambda is distributed asymptotically (under the null hypothesis of zero PDC) as the weighted sum of two chi-square with

one degree of freedom (Takahashi et al., 2007). Therefore, we can use this asymptotic distribution to calculate the p-values. For details of the derivation, we refer to Takahashi et al. (2007). Significance levels for frequency domain quantities are controled only point-wise as this is the standard everywhere. The reason for this is that the point estimates for neighboring frequencies are highly correlated. Therefore, standard correction like bonferroni or even FDR that assume independence or weak dependence give the wrong signficance level. Every single article that we found where PDC, coherence or bi-coherence were used and the signficance level is reported use the frequency-wise significance level (for representative examples see Huybers and Curry, 2006 and Came et al., 2007). For PDC it is easy to see that the use of frequency-wise significance level is reasonable given that the PDC values for different frequencies are the fourier transform of the same coefficents of the autoregressive process. The fact that lower frequency have fewer samples are taken care by higher threshold values for PDC at lower frequencies. We added the following brief description of the statistics for PDC in the main text. "PDC is a function of the coefficients of vector autoregressive model. Given that the coeffcients are asymptotically jointly normally distributed, we can use the delta method (Serfling, 1980) to obtain analytically the asymptotic statistics for PDC. After na algebraic computation we can show that PDC at frequency lambda is distributed asymptotically (under the null hypothesis of zero PDC) as the weighted sum of two chi-square with one degree of freedom (Takahashi et al., 2007). Therefore, we can use the the asymptotic distribution to calculate the p-value. For details of the derivation, we refer to Takahashi et al. (2007). The significance level used in the article for PDC is the frequency-wise value as it is the standard for frequency domain analysis given the high correlation between the point estimates for neighboring frequencies (see e.g. Huybers and Curry, 2006; Came et al., 2007)."

" 7) There is a recent paper: Franzke, C., D. Jelic, S. Lee and S. Feldstein, 2019: Systematic Decomposition of the MJO and its Northern Hemispheric Extra-Tropical Response into Rossby and Inertio-Gravity Components. Q. J. Roy. Meteorol. Soc., 145, 1147-1164. They use a composite approach which might be better suited to

investigate the MJO and QBO. Using linear regression might mix too many non-events into the analysis. C2 ESDD Interactive comment Printer-friendly version Discussion paper.

R:In the present version of the manuscript we have included an analysis derived from the reference suggest by the referee "Systematic Decomposition of the MJO and its Northern Hemispheric Extra-Tropical Response into Rossby and Inertio-Gravity Components. Q. J. Roy. Meteorol. Soc., 145, 1147-1164.". We believe that this analysis has lead to a better understanding of how the QBO affects each normal mode component of the MJO (ROT and Kelvin). In what follows we include the corresponding figures with corresponding descriptions.

Figure 2.7.1: MJO phase diagram showing all points (days) in which ãĂŰ(RMMãĂŮ_1ˆ2+ãĂŰRMMãĂŮ_1ˆ2≥1). Points marked in red

In order to exclude the cases in which the RMM index is not associated with a MJO event we excluded all cases in which ãĂŰ(RMMãĂŮ_1ˆ2+ãĂŰRMMãĂŮ_1ˆ2<1).,, among those cases we separated the ones for which the stratospheric zonal wind at 30Mb was positive (red) and negative (blue) in the figure 2.7.1. The MJO phase diagram was divided into 8 phases as in Franzke et. al 2019. For which QBO (positive or negative) state and for which MJO phase (i=1,2,...,8) we calculated the mean velocity and pressure fields associated with ROT and Kelvin modes at 200 Mb. Figures 2.7.2 and 2.7.3 are display respectively the composites associated with the reconstructions of velocity and geopotential height fields associated with ROT modes for each of the 8 MJO phases with positive stratospheric zonal wind at 30 mb (SZW30+) and negative (SZW30+). In order to compare both composites we compute the difference between SZW30+ and SZW30- of each field for each MJO phase. This is displayed in figure 2.7.4. We notice that for phases 1-3 the difference (of the geopotential height fields represented by the hatched region) is statistically significant for almost the entire domain. For phase 4 the fields are more similar with small regions with significant difference, associated with Rossby double vortices. Between phases 5-8 the areas with

significant difference become larger again.

Figure 2.7.2: Reconstruction of the velocity and geopotential height fields associated with ROT modes with SZW30+ at 200 Mb.

Figure 2.7.3: Reconstruction of the velocity and geopotential height fields associated with ROT modes with SZW30- at 200 Mb.

Figure 2.7.4: Difference between of the velocity and geopotential height fields associated with ROT modes with SZW30+ and SZW30-. The hatched region corresponds to significant difference of the geopotential height values under 5% confidence level.

Figure 2.7.5: Reconstruction of the velocity and geopotential height fields associated with Kelvin modes with SZW30+ at 200 Mb.

Figure 2.7.6: Reconstruction of the velocity and geopotential height fields associated with Kelvin modes with SZW30- at 200 Mb.

Figures 2.7.5 and 2.7.6 display respectively the composites associated with the reconstructions of velocity and geopotential height fields associated with the Kelvin mode for each of the 8 MJO phases with positive stratospheric zonal wind at 30 mb (SZW30+) and negative (SZW30+). In order to compare both composites we compute the difference between SZW30+ and SZW30- of each field for each MJO phase. This is displayed in figure 2.7.7. We notice that for phases 1-3 the difference (of the geopotential height fields represented by the hatched region) is statistically significant for almost the entire domain. Unlike in the case of ROT modes, for the Kelvin modes the distribution of statistically significant difference is more even throughout a MJO cycle with a larger area on phase 2 and more similar fields on phase 4. It is possible to notice a propagation pattern with negative geopotential height anomaly beginning at phase 4 and ending at phase 7.

Figure 2.7.7: Difference between of the velocity and geopotential height fields associated with Kelvin modes with SZW30+ and SZW30-. The hatched region corresponds

to significant difference of the geopotential height values under 5% confidence level.

8) Please correct "Frankze" to "Franzke" in the references. The correction was made.

Please also note the supplement to this comment:
https://esd.copernicus.org/preprints/esd-2020-45/esd-2020-45-AC3-supplement.pdf
* * *

---

## Author Comment (AC4) · 24 Sep 2020

Due to the large number of figures the complete response is found in the attatched pdf file.

1) The analysis and interpretation of section 3 is suspect (and possibly in other sections). Figure 1 clearly shows regular and artificial peaks at regular (frequency) intervals most likely resulting from the bandpass pre-processing of the data. The features look similar to those which would appear in data convolved with a square filter. I recommend that suitable prefiltering is done to minimise these numerical artefacts (i.e. using appropriate tapering methods). The reviewer is correct that the signal was rectified to

[Figure]

analyze the effect oonly on the amplitude of the time series. Nevertheless, it seems that this procedure created some doubt about the validity of our result. Therefore, we re analyze the data without rectifying the the signal and now report this result. We apologize for this confusion. The new figure is the following.

Figure 3.1.1: PDC analysis of the RMM index, QBO and ozone at the fast (intraannual time-scale).

Figure 3.1.2: PDC analysis of the RMM index, QBO and ozone at the slow (inter-annual time-scale).

2) It seems to this reviewer that the annual cycle has been retained in the data. Presumably retention of the annual cycle and sub-harmonics will obscure attribution of causality between the various timeseries? Why has the annual cycle been retained and what impact will this have on the interpretation of the results? We did not remove the annual cycle. The reviewer is correct to mention that the annual cycle is a dominant component of the all spectra investigated here, this however is not a problem once other spectral peaks of interest (i.e intraseazonal, biennial, interannual and decadal) are well represented by the parametric spectral estimation procedure. As explained in answer to question 7 of this reviewer our ability to well represent the spectral peaks of interest rely on the order of the auto-regressive model of choice.

3) The authors should provide figures for the timeseries used in the paper, before and after processing, including those short and long timeseries used throughout the manuscript. Here we include a new figure with the corresponding time-series which will be included in the new version of the manuscript. We have included the following figure:

Figure 3.3.3: Time-series of the RMM index, stratospheric zonal wind at 30Mb and equatorial ozone.

4) The authors have not justified the use of indices thought relevant for MJO-QBO
connections, namely MJO indices and the westward propagating gravity wave modes (and various others wave modes). There are a number of competing mechanisms for explaining the observed correlations between the MJO and QBO. A number of these do not explicitly involve waves, but rather upper tropospheric temperature, wind-shear or static-stability. The title of the paper suggests a focus on waves, but this needs to naturally come following an appraisal of the various mechanisms first.

R:The study of QBO effects on the MJO gained a lot of interest in the last few years, since new evidence pointed out to this connection (see Yoo, C., & Son, S. W. (2016). Modulation of the boreal wintertime Madden‐Julian oscillation by the stratospheric quasi‐biennial oscillation. Geophysical Research Letters, 43(3), 1392-1398.). Since then several articles explored both the physical mechanisms behind this interaction as well as consequences to weather and climate. One of the main factors that plays a factor in the QBO-MJO connection is the difference in the static stability at the Tropopause region depending on the phase of the QBO (see Nishimoto, E., & Yoden, S. (2017). Influence of the stratospheric quasi-biennial oscillation on the Madden–Julian oscillation during austral summer. Journal of the Atmospheric Sciences, 74(4), 1105-1125.). Hendon et. al 2018 suggests that negative temperature anomalies at the tropopause region at the eastern QBO act act to destabilize the upper troposphere in phase with MJO associated convection, thus reinforcing the MJO event (see Hendon, H. H., & Abhik, S. (2018). Differences in vertical structure of the Madden‐Julian Oscillation associated with the quasi‐biennial oscillation. Geophysical Research Letters, 45(9), 4419-4428.). Alternative mechanisms that could contribute to this stratosphere-troposphere connection include the downward reflection of planetary waves (see Lu, H., Scaife, A. A., Marshall, G. J., Turner, J., & Gray, L. J. (2017). Downward wave reflection as a mechanism for the stratosphere–troposphere response to the 11-yr solar cycle. Journal of Climate, 30(7), 2395-2414.) and effects on tropospheric Rossby wave-guides and teleconnection patterns (see Feng, P. N., & Lin, H. (2019). Modulation of the MJO‐related teleconnections by the QBO. Journal of Geophysical Research: Atmospheres, 124(22), 12022-12033.). Here we investigate

a different class of mechanism, namely the role of wave interaction. Nonlinear wave interaction is believed to have a role in the initiation of an MJO event though the interaction between the tropics and extra-tropics (see section 6.4 of Khouider, B., Majda, A. J., & Stechmann, S. N. (2012). Climate science in the tropics: waves, vortices and PDEs. Nonlinearity, 26(1), R1.). This interaction takes place by by the coupling between equatorially confined modes, the baroblinic Rossby waves, and non-confined modes, the barotropic Roosby waves. Inspired by this type of mechanism we investigate whether the interaction between QBO-related modes with MJO-related modes could have a role in the MJO-QBO connection.

5)The various horizontal/vertical normal modes used to construct QBO and MJO patterns and timeseries need to be captured somewhere (e.g. supplementary materials) as they feature prominently in the analysis. R:In our analysis we have used no truncation on the zonal wave- number with K=32 and vertical index up to M=43. The selection of modes is made on the type of the mode (rotational or inertio-gravity). On the meridional index, for the MJO only the first three modes symmetric wind structure with respect to the equator (indices n=1,3,5) were used for the rotational mode and the Kelvin mode (eastward inertia-gravity with meridional index n=1).

6) There is a lot of various missing information on the figures (labels, units, tickmarks etc), which has mostly been identified in the points below. All figures need to be improved for future review. R:In the new version of the manuscript we have included corrected versions of the figures.

7) The spectra look very smooth; has any smoothing been applied to the power spectra? If so, how has this been achieved? R:Yes, the whole PDC analysis relies on a autoregressive estimation of the spectra, this means that the choice of the autoregressive order will determine the smoothness of the spectra. The lower the chosen model less spectral peaks will be captured by the parametric estimate of the spectra, meaning that only the dominant spectral peaks will be represented, conversely high order models will be able to capture the fine structure of the spectra. In our analysis the order of

the autoregressive fitting was in the range 10-15, and were well adjusted according to the Portmanteau test. This means that the resulting spectra will be fairly smooth.

8) Figures 8-11. What physical mechanism will causally link wave modes on interannual to decadal timescales? What hypothesis is being tested? R:In our analysis we have calculated the energy time-series associated with normal modes and tested the causality between these energy time-series. We regard this as an evidence for nonlinear wave interaction similar to the barotropic-baroclinic Rossby wave interaction that plays a role in the initiation of the MJO (see Majda, A. J., & Biello, J. A. (2003). The nonlinear interaction of barotropic and equatorial baroclinic Rossby waves. Journal of the atmospheric sciences, 60(15), 1809-1821.).

9) Can the authors put forward a plausible physical mechanism linking the Himalayas near 30-40N and two equatorially confined phenomena – MJO and QBO? Furthermore, how should this mediate the observed statistical relationship between the QBO and MJO?

R:In the present version of the manuscript we have removed this section of the article and replaced it by a composite analysis showing the evolution of each normal mode component of the MJO following the suggestion of Reviewer #2. However, the idea here is that the strong divergence associated with these topographic gravity waves would act as a source of barotropic (in the troposphere) Rossby waves that could interact with the MJO via tropical-extra tropical interaction. We however acknowledge that this is still highly speculative and think that the composite analysis brings much more information on the process.

10) The authors have looked at large scale circulation processes in assessing longtime scale relationships between the QBO and MJO. What though are the roles for small-scale gravity waves in linking QBO and MJO connections?

R:One of the possible roles of small scale gravity waves is related with their vertical propagation, which is known to be a major mechanism for the QBO, therefore differences on the vertical wave propagation could in principle affect both the QBO and tropical convection (associated with the MJO) (see Piani, C., Durran, D., Alexander, M. J., & Holton, J. R. (2000). A numerical study of three-dimensional gravity waves triggered by deep tropical convection and their role in the dynamics of the QBO. Journal of the atmospheric sciences, 57(22), 3689-3702.).

Please also note the supplement to this comment:
https://esd.copernicus.org/preprints/esd-2020-45/esd-2020-45-AC4-supplement.pdf

---

## Author Response (AR2)

Cover Letter

Dear Mr. Editor,

We are hereby submitting the final version of the manuscript under the title "Stratospheric ozone and QBO interaction with the tropical troposphere on intraseasonal and interannual time-scales: a normal mode perspective". In this version of the manuscript we have addressed the remaining issues raised by the editor and by two of the reviewers, namely:

(1) We have expanded the section 2.3 of the manuscript which concerns the statistics of the partial directed coherence method in order to make it more self contained.

(2) We have revised the text correcting the typos and other minor errors.

We also changed the order of the authors, which was agreed by all authors.

We finally would like to thank both the editor and the reviewers for the excellent review process that lead to substantial improvement of our work.

Yours sincerely,

Breno Raphaldini